# A high-quality Data Set for seismological studies in the East Anatolian Fault Zone, Türkiye

Leonardo Colavitti[1], Dino Bindi[2], Gabriele Tarchini[1,3], Davide Scafidi[1], Matteo Picozzi[3,4], Daniele Spallarossa[1]

[1]University of Genoa, Department of Earth, Environmental and Life Sciences - DISTAV, Genoa, Italy
[2]German Research Center for Geoscience - GFZ, Potsdam, Germany
[3]National Institute of Oceanography and Applied Geophysics - OGS, Trieste, Italy
[4]University of Naples Federico II, Physics Department "Ettore Pancini", Naples, Italy

*Correspondence to*: Leonardo Colavitti (leonardo.colavitti@edu.unige.it)

**Abstract.** This work aims to develop and share a high-quality seismic data set for the East Anatolian Fault Zone (EAFZ), a highly active seismic area that is prone to earthquakes, as evidenced by the two major earthquakes of magnitude 7.8 and 7.6 that occurred on February 6, 2023 in central Türkiye and northern and western Syria.

The data set described here (available at https://doi.org/10.5281/zenodo.13838992, Colavitti et al., 2024) encompasses seismic
events from January 1, 2019, to February 29, 2024, focusing on small-to-moderate earthquakes from $M_L$ between 2.0 and 5.5 and is intended as a useful tool for researchers working on seismic source characterization and strong motion parameters.

The data set consists of 9,442 events recorded by 271 stations and includes a total of 270,704 seismic phases (148,223 P and 122,481 S). The Complete Automatic Seismic Processor (CASP) software package ensures accurate arrival times and refined earthquake locations, while the local magnitude is calibrated using a non-parametric approach. In addition to the earthquake
catalog, the data set features strong motion parameters such as selected Peak Ground Acceleration (PGA), Peak Ground Velocity (PGV), as well as Fourier Amplitude Spectra (FAS) in the frequency range from 0.05 to 47.2 Hz.

The disseminated product aims to support applications in spectral decomposition using the Generalized Inversion Technique (GIT), promote investigations in Local Earthquake Tomography (LET) and contribute to the development of Ground Motion Prediction Equations (GMPEs). Long-term objectives include studying the spatio-temporal evolution of seismicity to identify
preparatory processes for significant earthquakes, integrating this data with geodetic investigations, and enhancing earthquake hazard assessments.

# 1 Introduction

On February 6, 2023 a seismic sequence hit southern and central Türkiye and northern and western Syria along the East Anatolian Fault Zone (EAFZ). The sequence was started by a moment magnitude Mw 7.8 earthquake along the Nurdag-Pazarcık fault and followed, about 9 hours later, by a Mw 7.6 earthquake occurred north-northeast from the first shock, in Kahramanmaraş province, involving the Sürgü and Çardak faults (Güvercin et al., 2022; Dal Zilio and Ampuero, 2023; Melgar et al., 2023; Petersen et al., 2023). According to the Disaster and Emergency Management Authority (*Afet ve Acil Durum Yönetimi Başkanlığı,* AFAD), the combined death contribution in Türkiye and Syria exceeds 60,000 people, with more than 120,000 injured and an amount of economic losses of 163.6 billion USD, representing the deadliest natural disaster in the modern history of Türkiye at present, since the 526 Antioch event (Sbeinati et al., 2005).

In this study, we focus on the EAFZ, which is a portion of a major fault zone that runs through eastern Türkiye as it accommodates the tectonic relative motion between the Arabian and Anatolian microplates (Ambraseys, 1989). According to Melgar et al. (2023), the first event nucleated on a previously unmapped fault before transitioning to the East Anatolian Fault, which ruptured over a length of approximately 350 km, while the second one ruptured the Sürgü fault for ~160 km.

The goal of this study is to describe the procedure that led to the creation of a high-quality seismic data set for the EAFZ where the 2023 Kahramanmaraş event occurred and its dissemination to the scientific community to promote high-quality seismological research. Indeed, the development of high-quality data sets is crucial for the investigation of critical open issues, such as the estimation of source parameters (e.g. the seismic energy and the stress drop), which represent fundamental information for understanding fault mechanics and obtaining rupture scenarios for seismic risk mitigation, but are difficult to estimate and affected by large uncertainties (Cotton et al., 2013; Abercrombie, 2015).

Recently, benchmark studies have been carried out to facilitate comparison of the results of different approaches to estimate source parameters applied to the same data set (e.g., Pennington et al., 2021; Shible et al., 2022; Morasca et al., 2022; Bindi et al. 2023a; Bindi et al. 2023b). Following these efforts, a data set for the 2019 Ridgecrest earthquake sequence was disseminated in the context of the community stress-drop validation study (Baltay et al., 2024).

We believe that the creation of high-quality, standardized and open-source seismic data sets including waveforms, Fourier Amplitude Spectra (FAS), Peak Ground Acceleration (PGA), and Peak Ground Velocity (PGV), is the key to promote the progress of the seismological and seismic engineering communities. In this paper, we describe in detail the procedures used to construct the data set and the criteria applied for selecting the data to be distributed.

The data set includes earthquakes that occurred along the EAFZ main segments (**Fig.1**) in the period from January 1, 2019 to February 29, 2024, and thus includes both the years preceding the 2023 Kahramanmaraş earthquake and the following aftershocks. The data set focuses on small-to-moderate earthquakes in the magnitude range from 2.0 to 5.5, which is typically used in studies focusing on source parameters (Parolai et al., 2000; Parolai et al., 2007; Picozzi et al., 2017). Larger earthquakes are not included (besides the 2 mainshocks of February 6, 2023, we did not consider other 22 events with magnitudes from 5.6

to 6.6) as they are already available in accelerometric databases such as the Engineering Strong-Motion Database (ESM) by
Luzi et al. (2020) or in a recent work by Sandıkkaya et al. (2024).

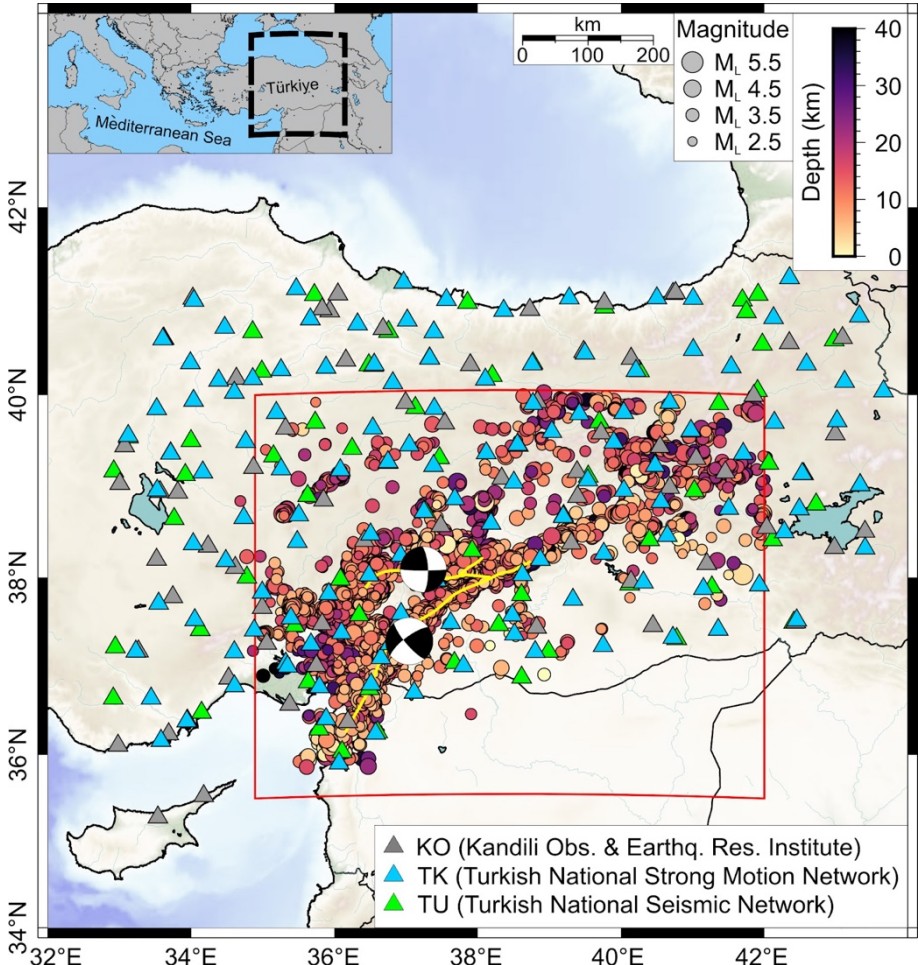

**Figure 1: The data set for the East Anatolian Fault Zone (EAFZ) described in this work, bounded by the red box (Lon-Lat vertices, SW: 34.89 35.50; SE 42.00 35.50; NE 42.00 40.00; NW: 34.89 40.0). Dots represent the events with $M_L$ between 2.0 and 5.5 in the period 01-01-2019 to 29-02-2024. The size is proportional to the magnitude; the color palette represents the event depth. The two beach balls lying in the Melgar faults (yellow lines) represent the Mw 7.8 Pazarcık earthquake and the Mw 7.6 Elbistan earthquake occurred on February 6, 2023, which are not considered in this catalog. The triangles show the different networks that recorded the events: KO (gray), TK (cyan) and TU (green).**

The distributed data set includes a selected seismic catalog, selected Peak Ground Acceleration (PGA), Peak Ground Velocity (PGV), and selected Fourier Amplitude Spectra (FAS) in the frequency range of 0.05-47.20 Hz.

The primary focus of the 2019-2024 EAFZ data set that we envision are the source parameters applications discussed previously. However, we believe it is particularly suitable for investigating the evolution of ground shaking patterns in space and time during seismic sequences.

Moreover, in the light of recent studies (Picozzi et al., 2022; Picozzi et al., 2023a; Picozzi et al., 2023b) on spatio-temporal analysis of seismicity and ground motion parameters (i.e., GMA - Ground Motion Anomalies - defined in Picozzi et al. 2024), the provided data set can support seismic studies for intercepting the preparatory phase of strong earthquakes.

**2 Data set construction and selection**

The flowchart in **Fig. 2** shows the process that led to the creation and selection of the data set. We used the AFAD online
catalog to geographically select all earthquakes that occurred between 32 and 44° East longitude and 34 to 43° North latitude (at this stage considering an area larger than the only EAFZ bounded by the red rectangle in **Fig.1**), at a depth of up to 120 km, a $M_L$ in the range 2.0-5.5 and for the period from January 1, 2019 to February 29, 2024. The initially selected reference catalog consists of 78,728 events, which are shown in the map in **Fig. S1** of the Supplement.

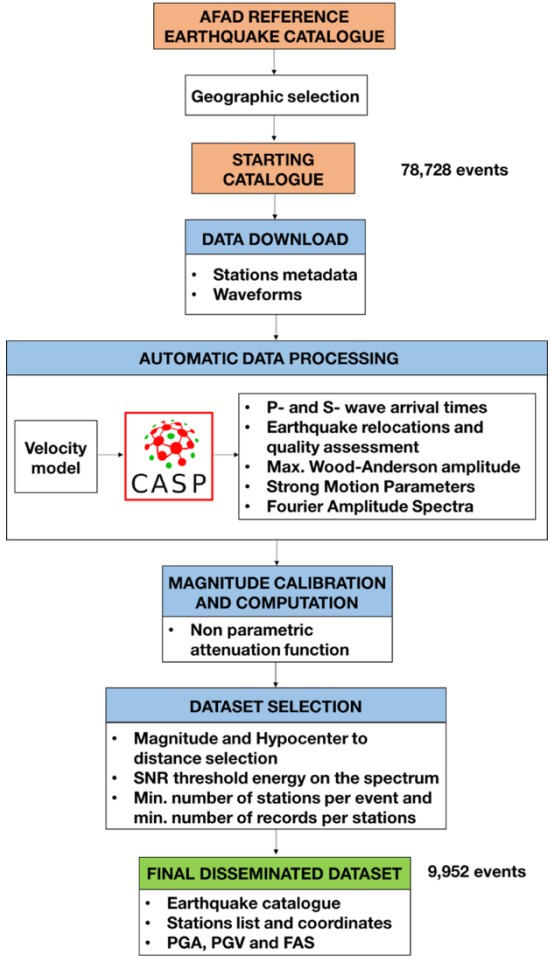

**Figure 2: Flowchart of the approach adopted in this work for the creation of the data set. Red boxes represent the catalogs of the data set, blue boxes the main procedure of the calculations. Acronym abbreviation: CASP (Complete Automatic Seismic Processor). The last box represents the disseminated data set discussed in this work.**

The starting catalog was downloaded through the International Federation of Digital Seismograph Networks (FDSN, https://www.fdsn.org/) web service, using the *fdsnws-event* command to access event parameters via the AFAD repository (reference website: https://deprem.afad.gov.tr/event-catalog). We downloaded the metadata for the stations belonging to the KO (Kandilli Observatory And Earthquake Research Institute, Boğaziçi University, 1971), TK (Disaster and Emergency Management Authority, 1973) and TU (Disaster and Emergency Management Authority, 1990) networks implemented in the data centers of AFAD and European Integrated Data Archive (EIDA, https://www.orfeus-eu.org/data/eida/, last accessed on February 29, 2024).

All waveforms for the three ground motion components were downloaded in MiniSEED format from the EIDA and AFAD repositories using the *fdsnws-dataselect* command. The seismograms of the events were extracted by selecting segments from continuous seismic recordings and converting them into the Seismic Analysis Code (SAC) format. Each time window contains 30 seconds of noise before the theoretical P-wave first arrival and has a total duration of 90 seconds. The entire earthquake catalog, with around 78,000 events, contains waveforms of different quality.

While studies focusing on statistical seismology (e.g., deviations from the Gutenberg-Richter law, such as b-value studies) are sometimes less sensitive to certain aspects of data quality, ensuring high data quality is critical for the accurate derivation of source parameters and the calibration of ground motion models, making the implementation of thorough data selection and quality analysis procedures a priority.

Therefore, to generate a high-quality data set, which is the most innovative aspect of this work, we used the Complete Automatic Seismic Processor (CASP, Scafidi et al., 2019) software, which determines seismic phase arrival times with an advanced picker engine (RSNI-Picker$_2$, see Spallarossa et al., 2014; Scafidi et al., 2016; Scafidi et al., 2018). This process resulted in a massive set of accurate P- and S-wave arrival times consistent with earthquake locations. RSNI-Picker$_2$ provides a quality estimate for each computed parameter, such as the automatic pick weighting and standard location quality metrics.

The search of reliable seismic phases arrival times in CASP is linked to and driven by seismic locations. To obtain reliable seismic locations, the Non-Linear Location (NLLoc, Lomax et al., 2000; Lomax et al., 2012) algorithm was used, which implements a regional velocity model specifically suited for the EAFZ (Güvercin et al., 2023). As mentioned in Spallarossa et al. (2021a), CASP enhances detectability by improving the accuracy of arrival times, increasing reliability, minimizing the rate of false picks, and, in general, the accuracy of results. The final result of the CASP procedure is a data set of P- and S-phase arrival times and an earthquake catalog of origin time, location, depth and local magnitude $M_L$, all seamlessly linked together.

In the initial processing phase, $M_L$ was calculated using a generic calibration relationship (Hutton and Boore, 1987). After processing the selected 2019-2024 EAFZ data set with CASP, a new calibration relationship was developed using a non-parametric approach (see **Section 3.2**), and magnitudes were recalculated for all events.

In this work, the automatic procedures of the CASP software also provide some strong motion parameters such as PGA, PGV and Fourier Amplitude Spectra (FAS).

The method used to calculate the FAS and to select the dataset is described in detail in Pacor et al. (2016) and has been applied in subsequent studies (e.g., Picozzi et al., 2022; Castro et al., 2022a) to analyze seismic sequences in Central Italy.

The FAS were calculated considering 98 frequencies equally spaced on the logarithmic scale in the frequency range 0.05-47.2 Hz and smoothed using the Konno and Ohmachi (1998) algorithm, with the smoothing parameter b set to 40.

The selection of the high-quality data set is therefore based on the following criteria:

(i)      Events restricted to the EAFZ (see red box in **Fig. 1**, Lon-Lat vertices, SW: 34.89 35.50; SE 42.00 35.50; NE 42.00 40.00; NW: 34.89 40.0);

(ii)      $M_L$ between 2.0 and 5.5;

(iii)      Hypocentral distance up to 150 km;

(iv)      A recursive procedure ensuring that at least 60% of the Fourier spectra have a points that satisfy Signal to Noise Ratio (SNR) greater than 2.5;

(v)      Events recorded by at least 6 stations, with each station having a minimum of 6 recordings.


After selection, the final data set ready for distribution described in the following sections, consists of 9,442 events recorded by 271 stations, including 142 strong-motion (channel HN), 123 high-gain high-broadband (HH), 4 high-gain broadband (BH) and 2 short-period seismometers (EH). In total, the data set comprises 843,651 waveforms for the three ground-motion components.

The 2019-2024 EAFZ high-quality data set consists of an earthquake catalog, a table with the coordinates of the stations used and the values of strong motion parameter, such as PGA, PGV and FAS (see **Section 5 - Data availability** for more details).

### 3 Data set characteristics

As we can observe in **Fig. 3**, the data set is well sampled with approximately 50% of earthquakes recorded by 10 stations and about 50% of earthquakes having more than 100 records. More details on the distribution of the number of recordings per event and per station can be found in **Fig. S2**.

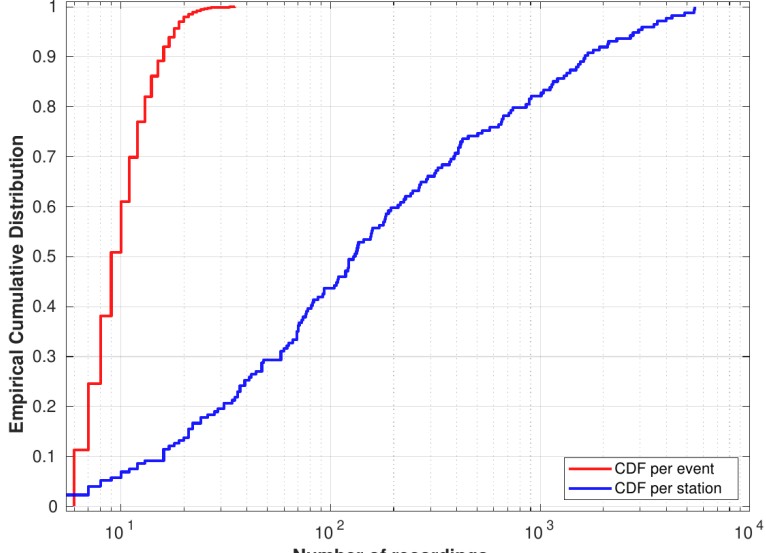

**Figure 3: Cumulative Distribution Function (CDF) per event (blue) and per station (red) used in the data set.**

**Figure 4** represents the heat map of the recordings in terms of hypocentral distance and $M_L$ at the sampling frequency of 1 Hz (variations of the heat map at different frequencies can be found in **Fig. S3**). The figure shows that the most sampled area is at about 70 km hypocentral distance and $M_L$ 3, with some cells (with a resolution of 4 km for hypocentral distance and 0.1 for magnitude) reaching up to 300 counts.

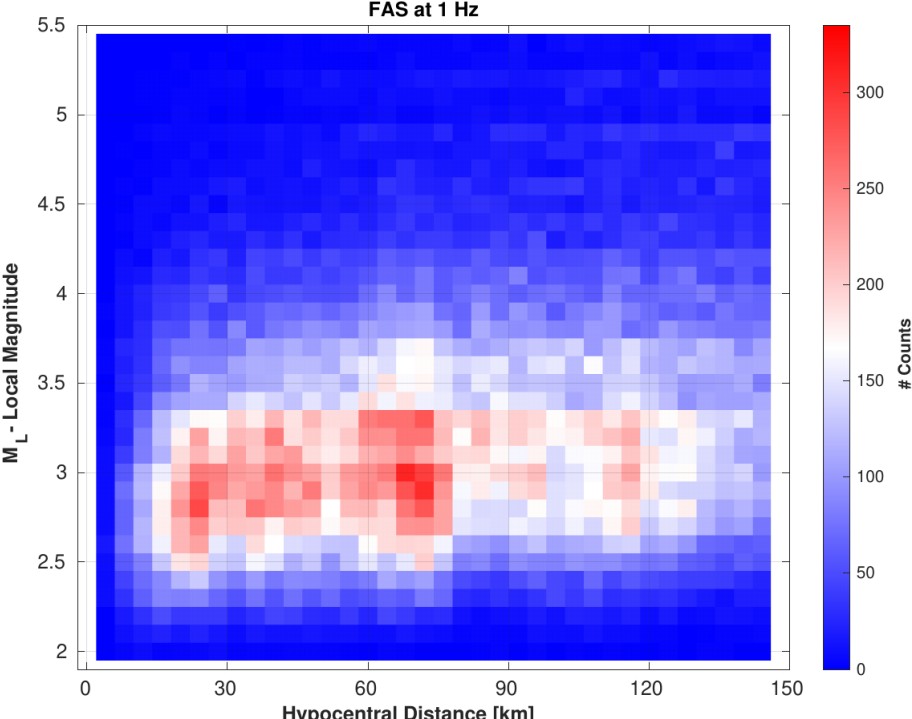


**Figure 4: Local magnitude versus hypocentral distance of the recordings considered in this study at FAS = 1 Hz.**

More than 50% of the records include a hypocentral distance < 80 km, about 67% a hypocentral distance of less than 100 km and more than 80% of the data are recorded at a hypocentral distance < 120 km. Regarding the magnitude distribution, about
50% of the observed recordings are below $M_L$ 3.2, about 67% below $M_L$ 3.5 and around 86% below $M_L$ 4.0.

### 3.1 P and S Phase picking and event relocation using CASP

The event relocation is performed using the Complete Automatic Seismic Processor procedure (CASP) which has been described in details in previous studies (Scafidi et al., 2018; Scafidi et al., 2019).

CASP consists of 4 main steps, which can be summarized as follows:

• Step 1 (RSNI-trigger module): the signal is band-pass filtered to retrieve P onsets: a preliminary Short-Term Average/Long-Term Average (STA/LTA) is calculated. If the ratio exceeds a threshold value, a single-station trigger is declared: the final trigger time is then determined as the minimum of the Akaike Information Criterion (AIC) functions (Akaike, 1974). The output of the RSNI-trigger module is a list of trigger times for each station.

• Step 2 (RSNI-detect module) is designed to be effectively applied to different cases, from small local seismic
networks to large and dense regional ones. The detector algorithm is based on a system defining the number of data channels that must have triggered within a coincidence window to declare the start of a potential event. The detector algorithm is based on a module that performs an additional check based on the comparison among earthquake

locations computed using the trigger times as P-phase picks by the NLLoc software (Lomax et al., 2000; Lomax et al., 2012), with a configuration optimized for the study area.

- Step 3 (RSNI-extract module): the seismograms of the detected potential events are extracted from the data set containing all continuous recordings and converted to SAC format.

- Step 4 (RSNI-picker$_2$ module): the extracted seismograms relevant to each recognized event are processed using the RSNI-picker$_2$ module (Spallarossa et al., 2014; Scafidi et al., 2018; Scafidi et al., 2019) to determine the P- and S-phase arrival times, the earthquake locations, to calibrate the magnitude and compute strong-motion parameters. The

first set of iterations for P- and S- phases is followed by a quality check of the location based on the number of phases and computed location errors, which controls the triggering of the second set of iterations for the S phases. The quality of the final solution is also assessed by considering time residuals and other predictors.

For the study area, the seismic locations were obtained using the EAFZ regional velocity model obtained by Local Earthquake Tomography by Güvercin et al. (2022), which is a multi-layer model using VELEST inversion code (Kissling et al., 1994)

obtained using 700 selected events with azimuthal gap < 80° and relocated using at least 25 phase readings.

Detailed information on the velocity structure used as the initial model, can be found in **Table S1**.

The distributed data set comprises a total of 270,704 phases selected by CASP (see **Fig. S4a**). **Figure 5** shows the distribution of the P and S phases.

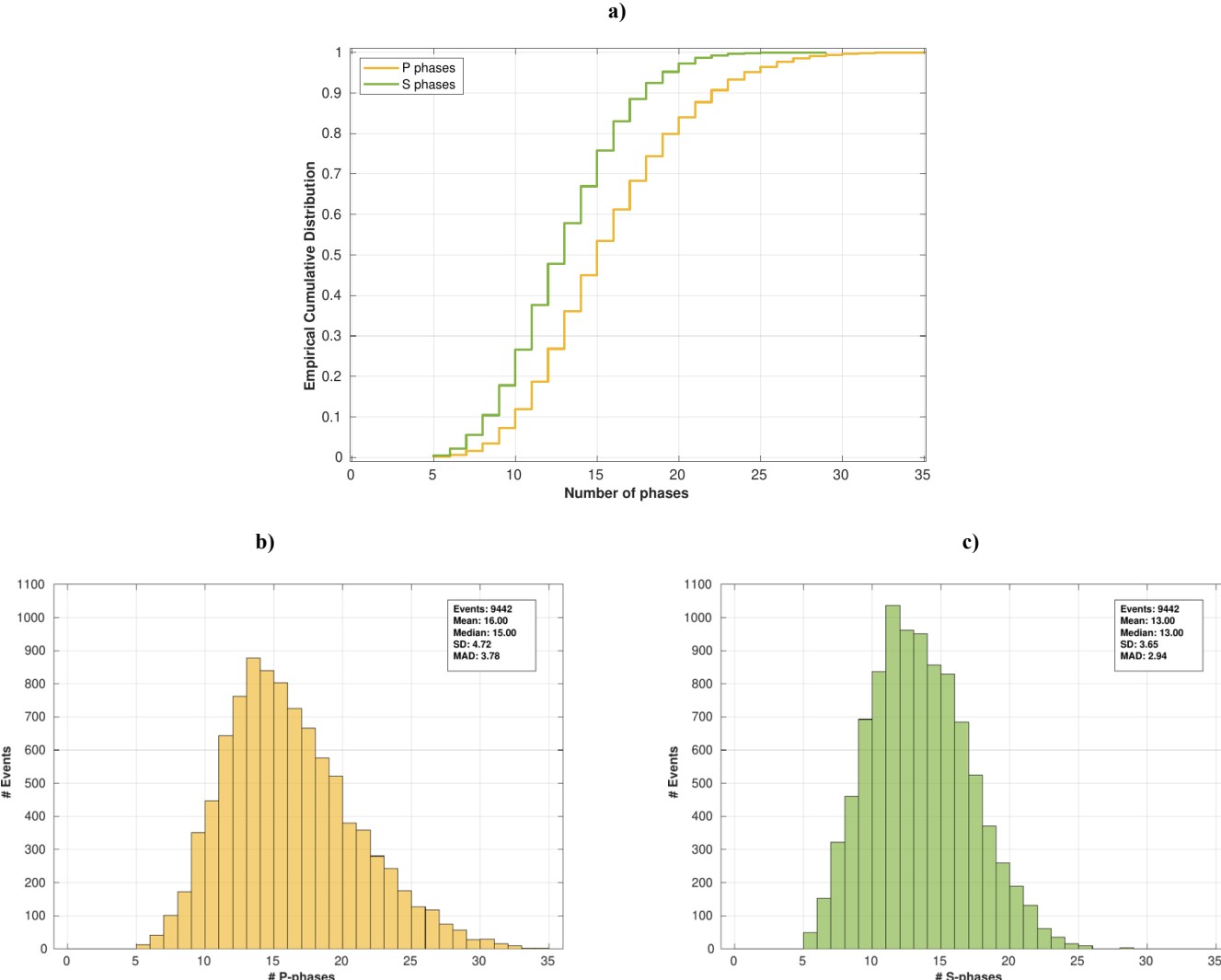

**Figure 5: a) Empirical Cumulative Distribution of P and S phases in the present data set. b) Histogram of number of P-waves, in yellow b) and S-waves, in green c) picked by the Complete Automatic Seismic Processor procedure (CASP).**

The minimum number of total seismic phases recorded per event is 10 (5 for the P-phase and 5 for the S-phase) and, as can be seen in **Fig. 5a**, 50% of the entire data set has at least 15 P-phases (yellow curve) and 13 S-phases (green curve). **Fig. 5b** and **Fig. 5c** show that the most frequent value in the distribution of histograms is 13 for the P-phases and 11 for the S-phases. A total of 148,223 P- and 122,481 S-phases were used, which corresponds to a ratio of approximately 55% to 45% of the total seismic phases considered. This considerable amount of high-precision P- and S-phase arrival times can be well utilized by the seismological community involved in tomographic studies, especially in the context of Local Earthquake Tomography (LET) investigations.

### 3.1.1 Quality of earthquake location

NLLoc provides the quality and uncertainty of seismic locations through several parameters, including horizontal and depth error. The horizontal error (Err H) is expressed as a confidence region in the horizontal plane, indicating the area where the earthquake is located with a certain probability, while the depth error (Err Z) is derived from the vertical axis of the error ellipsoid. **Fig. 6a** shows the empirical distribution curve for the location error.

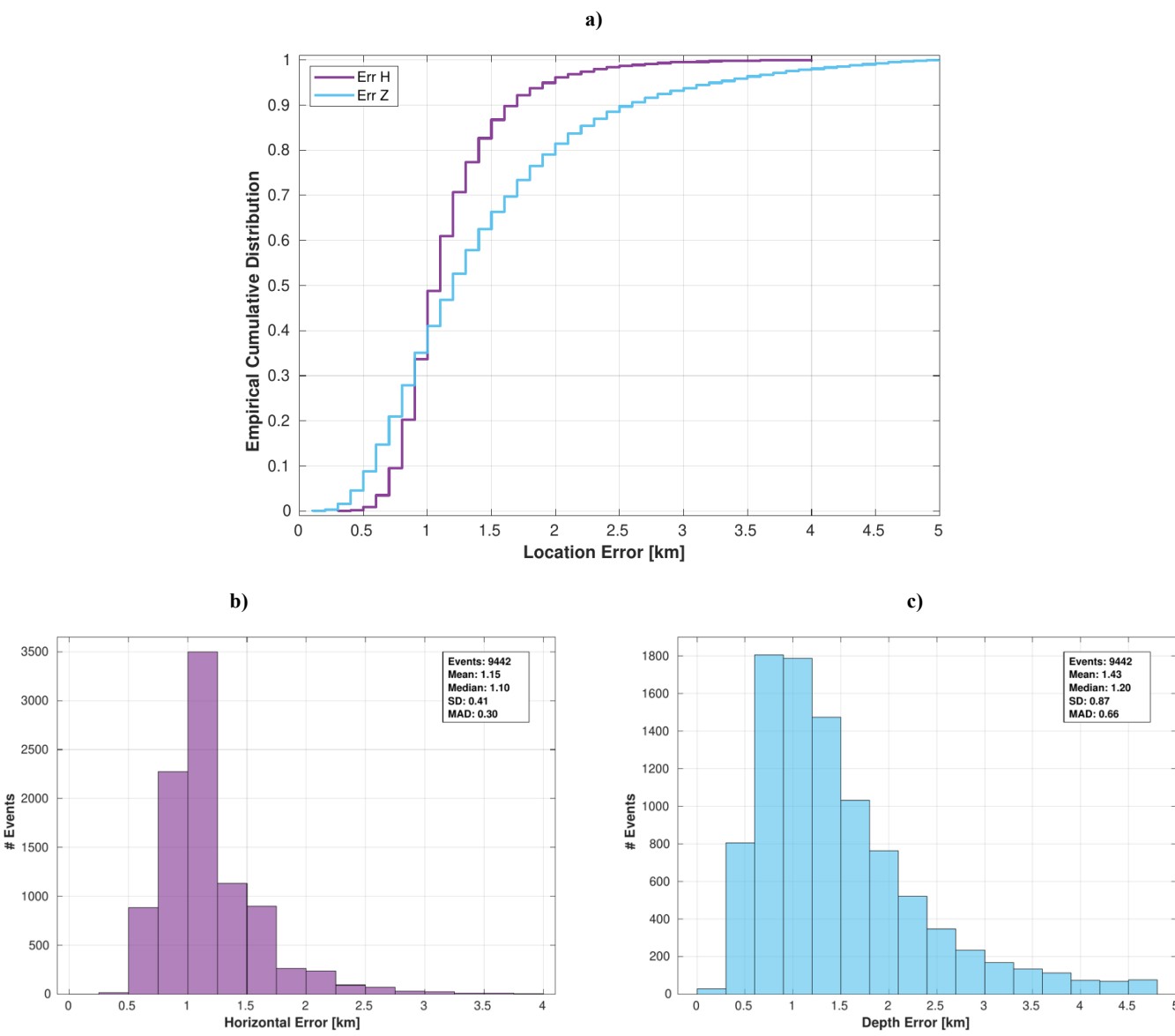

**Figure 6: a) Empirical Cumulative Distribution of horizontal error and error in depth in the present dataset. b) Histogram of horizontal error (purple) and depth error (cyan) provided by Non-Linear Location (NLLOC).**

The histograms of horizontal location error (**Fig. 6b**, in purple) and depth error (**Fig. 6c**, in cyan) show that the overall uncertainty is small and has a median error of 1.1 km for the horizontal location (min Err H: 0.30 km; max Err H: 4 km) and 1.2 km for the depth location (min Err H: 0.1 km; max Err Z: 5 km).

Both horizontal and depth errors are influenced by several factors, including seismic network geometry, travel-time measurement accuracy and velocity model complexity. In general, we can affirm that the NLLoc algorithm can determine the epicentral location with a precision of ±1 km, even in the presence of errors in crustal velocities, as observed in Laporte et al. (2024), who address uncertainties in earthquake location using different techniques derived from the Global Sensitivity Analysis (GSA) framework. A further indicator of the high quality of the seismic locations obtained is the azimuthal gap, which reflects how well the seismic stations are distributed around the earthquake location. Approximately 98.5% of the events have gaps of less than 180°, with over 70% of the events having a gap below than 90° and the median gap being 75° (see histogram on **Fig. S4b**). Root Mean Square (RMS) error is also a key parameter for assessing the earthquake location quality. The RMS is defined as the difference between the observed and calculated seismic wave arrival times at the stations. In formula:

$$RMS = \sqrt{\frac{1}{N}\sum_{i=1}^{N}\left(t_{obs,i} - t_{calc,i}\right)^2} \tag{1}$$

where:

- $t_{obs,i}$ is the observed arrival time at the station $i$;

- $t_{calc,i}$ is the computed arrival time at the station $i$;

- $N$ is the total number of stations that recorded the event.

Low RMS error values indicate good agreement between observed and calculated arrival times, suggesting accurate seismic location. Conversely, high RMS values indicate larger discrepancies and potential location errors. According to Lienert and Havskov (1995), an RMS value between 0 and 1 seconds indicates a highly accurate location. In this study (see **Fig. S4c** for details), the maximum RMS values reached 0.89 seconds with a median of 0.27 s.

A further parameter providing indications about the reliability of seismic location reliability in the NLLoc (Lomax, 2000) software is the covariance matrix, a square matrix describing the variance and covariance between residuals, which is solved for 4 unknowns: spatial coordinates x, y, depth z and time t. As provided in **Fig. S5** of the Supplement, which shows histograms for the covariance values, $Cov_x$ and $Cov_y$ are small, with a median around 0.5 and a mean around 0.75. In contrast, $Cov_z$ and $Cov_T$ exhibit higher values with median values of 1.5 and 1.9 and mean around 2.8 and 3.2, respectively.

### 3.1.2 Epicentral comparison with AFAD catalog

**Figure 7** compares event location of each event, computing the distance between the epicenters identified in this study and those given in the AFAD catalog. The distance is computed using the haversine formula (short for half versed sine, see Robusto, 1957), with the Earth radius fixed at 6371 km and the great circle distance between 2 points on a sphere based on their longitude and latitude.

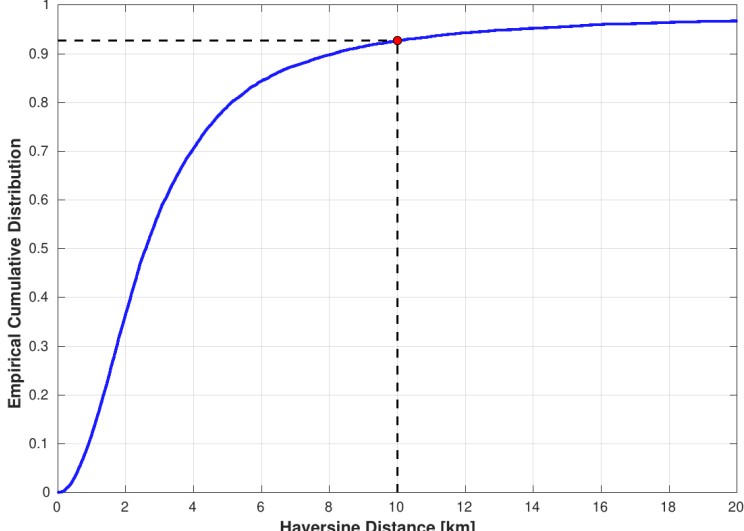

**Figure 7: Cumulative Distribution Function (CDF) with the Haversine distance computed between the location obtained in this work and the one provided by the AFAD catalog. Black dashed line shows the distance at 10 km, corresponding to the 92.7% of the CDF (blue curve); red point shows the intercept.**

The CDF shows that about 80% of the entire data set has a location difference within 5 km, 92.7% of the events are below 10 km (marked by the red intersection point) and over 97% of the events are within 20 km. This again confirms the reliability and precision of the epicenters in our catalog.

### 3.1.3 Depth comparison with AFAD catalog

We compared the hypocentral depth estimates obtained with NLLoc with those from the AFAD EAFZ catalog. The latter shows that most events are located at about 7 km (i.e., 80% of the depths are between 6 and 8 km). Such clustering in the hypocentral depth estimates often indicates the presence of a high gradient in the velocity model used in the inversion procedure. This might also be the case for the AFAD catalog, as discussed by Çıvgın and Scordilis (2019).

As shown in **Fig. 8a**, our study (red histogram) shows a more even distribution over different depths compared to the AFAD catalog (**Fig. 8b**).


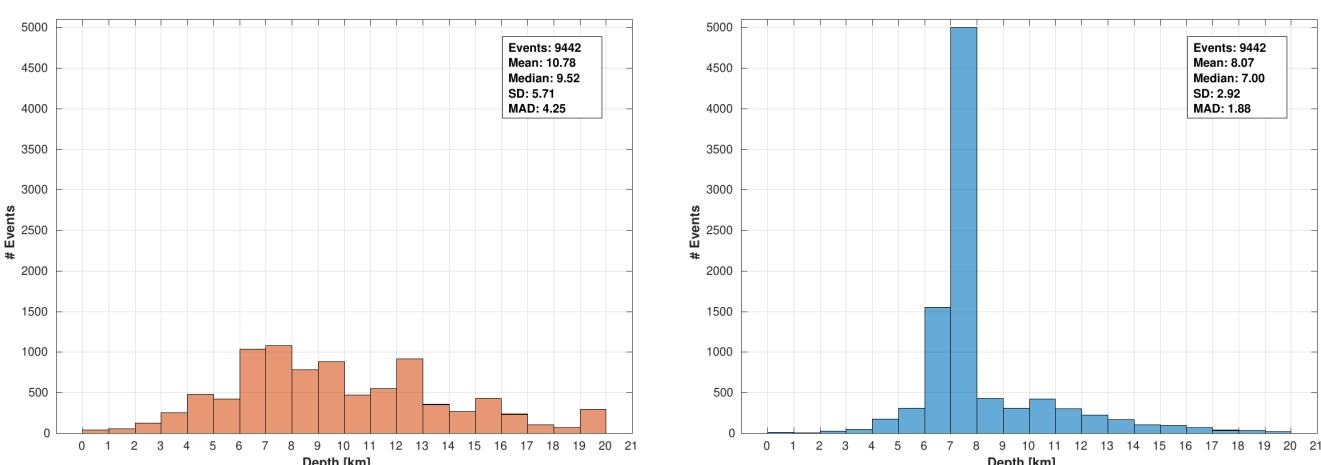

**Figure 8: Depth comparison between this work (a) and AFAD catalog (b).**

The median depth in this study is about 10 km, with no prominent peaks in the distribution. The relatively uniform depth estimates between 4 and 16 km appear to be consistent with the geometrical complexities of the fault segments in the 2023

Türkiye earthquake sequence (Gabriel et al., 2023).

**3.2 Magnitude computation**

To ensure a homogeneous magnitude for all considered earthquakes, we calibrated an $M_L$ (Richter, 1935) following a non-parametric approach (Savage and Anderson, 1995; Spallarossa et al., 2002; Bindi et al., 2018; Bindi et al., 2019; Bindi et al., 2020). Magnitudes were determined by applying station corrections to account for local site effects, and to avoid biasing the

amplitude measurements used for magnitude calculation. In our study (see **Fig. S6**), corrections of more than 0.60 had to be applied to 2 out of 271 stations (TK.6802 and TK.6102), as they are located at sites with low Vs,30 (less than 250 m/s). The non-parametric approach was applied to all data downloaded from the AFAD catalog.

With our analyses, we computed and provided the Wood-Anderson maximum amplitudes, which were then used to calibrate a EAFZ $\boldsymbol{M_L}$ scale based to the following equation:

$$logA_{ij}(R_{ij}) = M_{Li} + a_n logA_0(R_n) + a_{n+1} logA_0(R_{n+1}) + dM_{Lj}^C \tag{2}$$

where $A_{ij}$ is the maximum Wood-Anderson amplitude (in mm) measured for event $i$ recorded at the hypocentral distance $R_{ij}$. $M_{Li}$ is the local magnitude of event $i$, $A_0$ is the zero-magnitude attenuation function defined as a table of values $a_n$ linearly interpolated between nodes $n$ and $n+1$, with $R_n \leq R_{ij} \leq R_{n+1}$, while $dM_{Lj}^C$ is the magnitude correction of station $j$. $C$ can be

either North-South or East-West, considering the two horizontal components as independent measurements (Uhrhammer et

al., 2011). As for the reference distance at which the attenuation function is anchored, we used a distance $R_{ref}$ = 17 km (Hutton and Boore, 1987) and the mean value of all stations equal to zero. In **Fig. S7**, we show the output of the non-parametrically calibrated magnitude, with the comparison between the local magnitude $M_L$ and the magnitude of the AFAD reference catalog $M_{L\,ref}$.

**Figure 9** compares the calibrated $\log A_0$ function performed in the EAF (listed in **Table S2**) with the curve computed for the Southern California (Hutton and Boore, 1987).

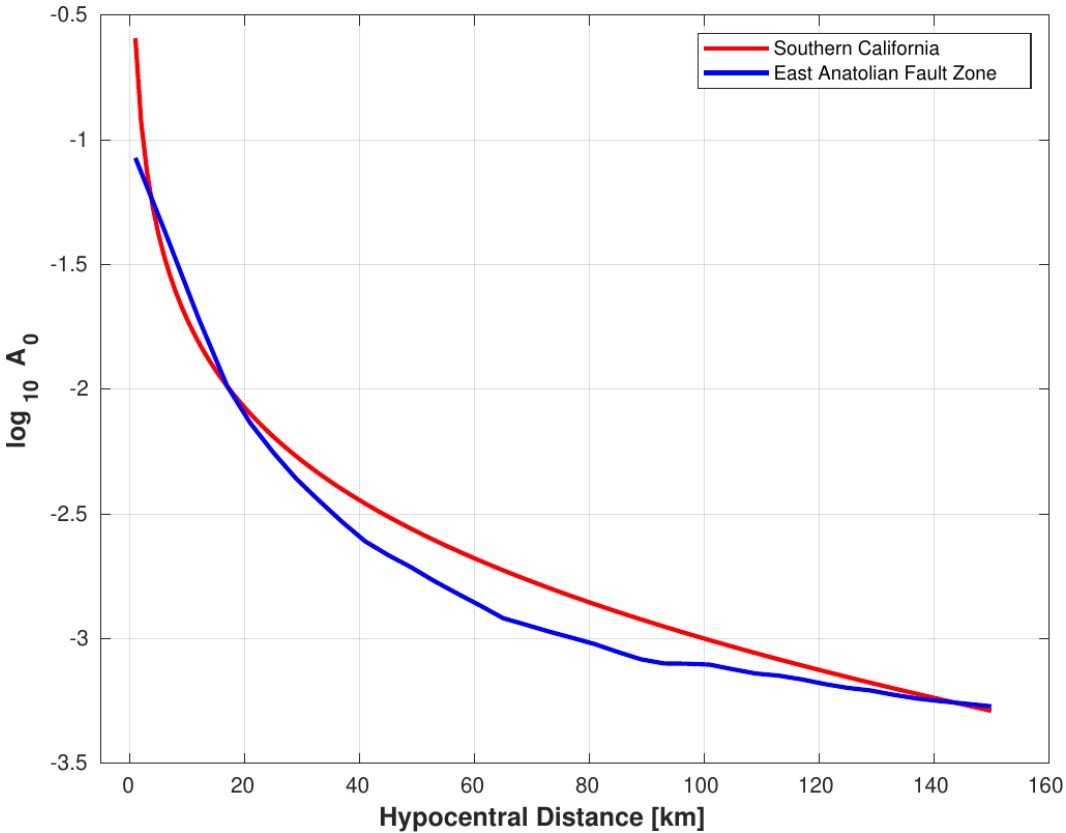

**Figure 9: Nonparametric magnitude attenuation function $\log A_0$ calibrated in East Anatolian Fault (blue curve) and Southern**
**California (red curve).**

In the distance range between 20 and 100 km, the EAF curve shows a stronger attenuation than the one computed for Southern California region, but is similar at hypocentral distances of more than 100 km. For each $M_L$ value, the standard deviation is also computed (**Fig. 10**). Both the mean and the median magnitudes of the analyzed events considered are around 3.1, with over 90% of the events having an $M_L$ value below 4 (**Fig. 10a**).


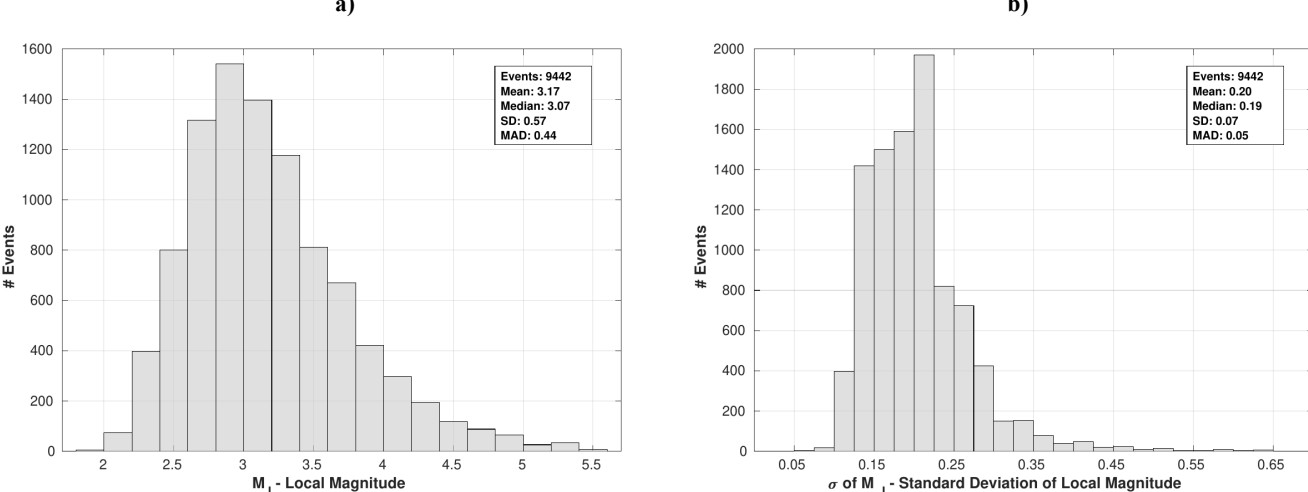

**Figure 10: a) Distribution of the local magnitude a) and the standard deviation b) of the events presented in this work.**

The standard deviation graph shows a median of 0.19 (mean 0.20) with generally low values, as less than 2% of the events have a magnitude uncertainty greater than 0.40, which shows that the $M_L$ measurement is very accurate.

We have also analyzed the cumulative frequency magnitude (CFM) distribution analyzing the *b*-value using the maximum likelihood approach (Aki, 1965) while the uncertainty of the *b*-value is computed by means of a bootstrap approach (Efron, 1979) and *b*-positive applying the procedure proposed by van der Elst (2021).

As written in **Figs. S8** and **S9** of the Supplement, the best estimates for the b-value is $0.89 \pm 0.01$ with *b*-positive $0.85 \pm 0.01$ considering all data set and $0.9 \pm 0.01$ for both *b*-value and *b*-positive considering the GIT distributed dataset.

### 3.2.1 Magnitude comparison with AFAD catalog

**Figure 11** shows the comparison between $M_L$ distribution the present data set (**Fig. 11a**) and with the magnitude of the AFAD catalog (**Fig. 11b**). It is worth mentioning that the AFAD catalog lacks a uniform parameter to characterize the earthquake size, so that other magnitudes types (e.g. duration magnitude, $M_D$ are used in addition to $M_L$).

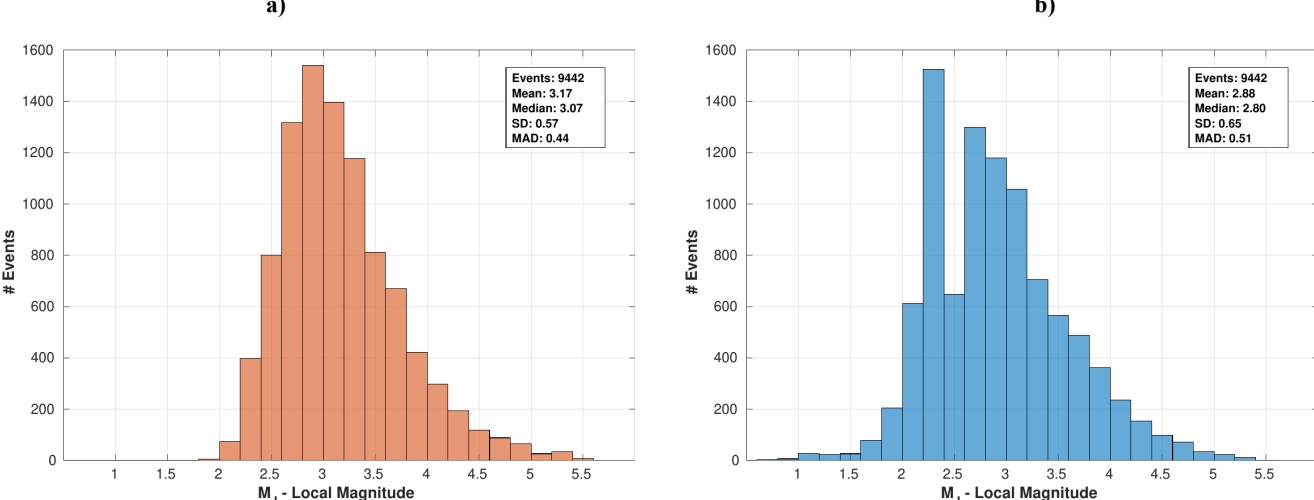

**a)**

**b)**

Figure 11a info box:
Events: 9442
Mean: 3.17
Median: 3.07
SD: 0.57
MAD: 0.44

Figure 11b info box:
Events: 9442
Mean: 2.88
Median: 2.80
SD: 0.65
MAD: 0.51

**Figure 11: Magnitude comparison between this work (a) and AFAD catalog (b).**

The magnitude distribution appears to be similar between the two data set. However, the median value in the AFAD catalog (around 2.8) is lower than in our dataset. Also, only a small part of the distribution, about 7%, includes magnitude values above 4, compared to 10% in our dataset.

**3.3 Strong motion parameters**

As mentioned before, we applied P and S onset detection to estimate $M_L$, and extract different several features from the recordings such as the peak displacement (PGD), the integral of the squared velocity (IVs2) evaluated over the S-wave window at local distances, the peak ground velocity (PGV) and the peak ground acceleration (PGA).

These features are extracted directly from the recordings and form the basis for the Rapid Assessment of MOmeNt and Energy

Service - RAMONES project (Spallarossa et al., 2021b, web page: https://distav.unige.it/rsni/ramones.php). This service provides seismic moment $M_0$ and radiated energy $E_r$, and relies on the measurement of specific ground motion features directly from seismograms and their correction for propagation and site effects using empirical models previously calibrated for the region of interest.

**Figure 12** shows an example of a three-component recording (N-S: North-South; E-W: East-West; Z: Vertical) relative to the

record $M_L$ 4.0 earthquake recorded at the station HASA occurred on March 20, 2023 at 15:40:34 UTC.

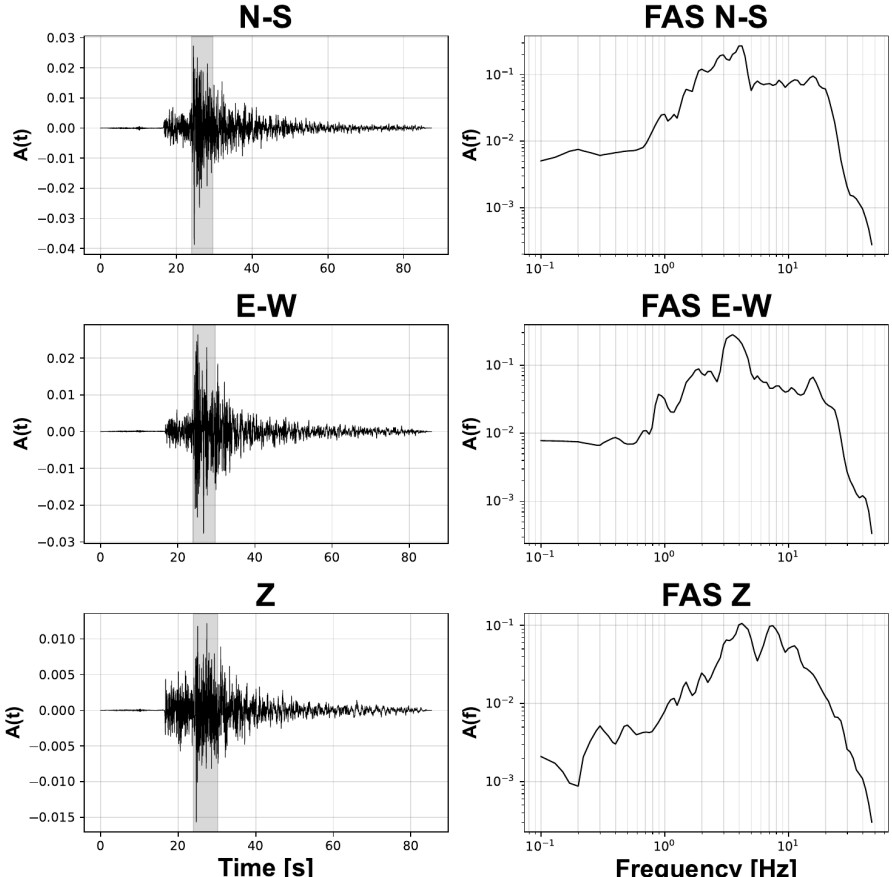

**Figure 12: Record for the M$_L$ 4.0 event occurred on 20 March 2023 at 15:40:34 UTC at the station HASA, where in gray is shown the portion of the signals for which FAS is computed. The panels on the left represent the seismic signals, on the right are the corresponding FAS. North-South component at the top, East-West component in the middle, vertical component at the bottom.**

Fourier Amplitude Spectra (FAS) are calculated for all three signal components using a recursive procedure based on a distance-dependent energy criterion to determine the S-wave time window length. A frequency-dependent threshold to the signal-to-noise ratio (SNR > 2.5) is then used to select the spectral amplitudes for the inversion (Pacor et al., 2016).

The FAS calculation is performed on time windows starting 0.1 s before the S-wave onset and extending until 60% of the total energy of the full spectrum is reached. The spectral amplitudes are calculated considering 98 frequencies, equally spaced in the logarithmic scale, in the range between 0.05 and 47.2 Hz. In addition to FAS, PGV and PGA values are also distributed as they provide a comprehensive overview of seismic motion and its potential impact on structures (Trifunac and Brady, 1975; Aki and Richards, 2002).

**Figure 13a** shows the values of log$_{10}$PGA as a function of hypocentral distance, defined for different magnitude ranges.

The solid line represents the median curve for a given magnitude, while the shaded areas indicate the variability within each of hypocenter bin distance, bounded by the lower quartile (25$^{th}$ percentile) and the upper quartile (75$^{th}$ percentile), respectively. PGA here refers to PHA (Peak Horizontal Acceleration), which represents the vector composition of the horizontal

components of strong ground motion:

$$PHA = \sqrt{(PGA_{N-S})^2 + (PGA_{E-W})^2} \tag{3}$$

where $PGA_{N-S}$ is the component of PGA along the North-South direction, while $PGA_{E-W}$ is the component of PGA along the

East-West direction.

The distribution of all events for which PGA is available,

The distribution of PGA and PGV with respect to the hypocentral distance can be found in **Figure S10**.


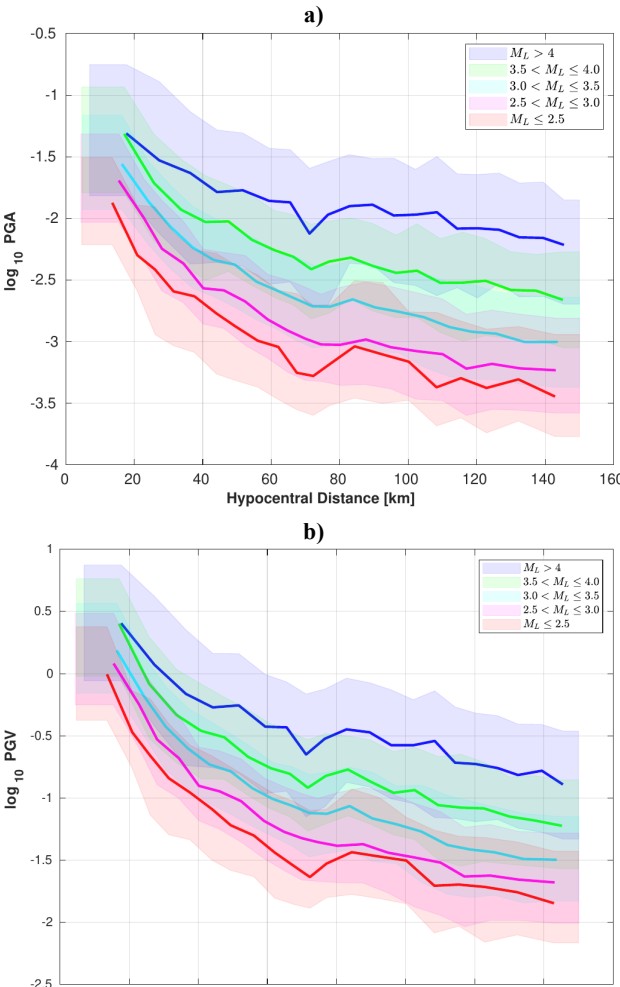

Figure 13: Distribution with hypocentral distance of the log10 PGA a) and log10 PGV b) over the S-wave window. The trends of the
parameters averaged over five narrow magnitude ranges as indicated in the panels ($M_L$<=2.5; $2.5 < M_L <= 3$; $3 < M_L <= 3.5$; $3.5 <
M_L <= 4.0$; $M_L > 4.0$) are also shown. Areas in transparency show the variability of each curve bounded by the lower quartile ($25^{th}$
percentile) and upper quartile ($75^{th}$ percentile).

As expected, $\log_{10}$PGA values are highest at small hypocentral distances, with median values between -2 and -1 at hypocentral
distances of less than 20 km. For intermediate hypocentral distances, between 20 and 80 km, $\log_{10}$PGA decreases, with median
values between -3 and -1.5. For larger hypocentral distances, beyond 80 km, $\log_{10}$PGA values appears to reach a plateau. A
similar trend is observed for PGV, as shown in **Fig. 13b**, in which the median $\log_{10}$PGV values are between -2 and 0.5. Curves
representing 5 magnitude ranges (red, $M_L$<=2.5; pink, $2.5 < M_L <= 3$; cyan, $3 < M_L <= 3.5$; green, $3.5 < M_L <= 4$; blue > 4.0)
show that higher $M_L$ values correspond to higher PGA. This effect is particularly evident at hypocentral distances greater than
40 km, where the curves are clearly separated by magnitude.




**4 Discussions: Seismicity distribution in the EAFZ**

Although the primary objective of this study is not a spatio-temporal investigation of seismicity in the EAFZ, we have mapped seismic activity from this catalog to obtain a general assessment of its distribution in relation to the major active geological structures. **Figure 14** shows a map of our study area illustrating seismicity over time, with the first mainshock on February 6, 2023 at 01:17:35 UTC serving as a reference point (time zero). Light green to green colors represent events that occurred further in the past (four and three years before the mainshock, respectively), while bluish to blue colors indicate events closer to the first mainshock. Events in the immediate vicinity of the first mainshock are shown in shades of red, whereas aftershocks that occurred at a later date are shown in shades of orange.

The map shows that seismicity is mainly concentrated along the southern Nurdağı-Pazarcık Fault, where the first event occurred, and near the Sürgü Fault, where the Mw 7.6 Ekinözü earthquake struck on February 6, 2023 at 10:24:49 UTC. This high-magnitude event, which occurred on a separate fault structure, is considered part of a "doublet" rather than a traditional mainshock-aftershock sequence (see Taymaz et al., 2022). To better visualize the seismicity distribution, we created several cross-sections.

**a)**

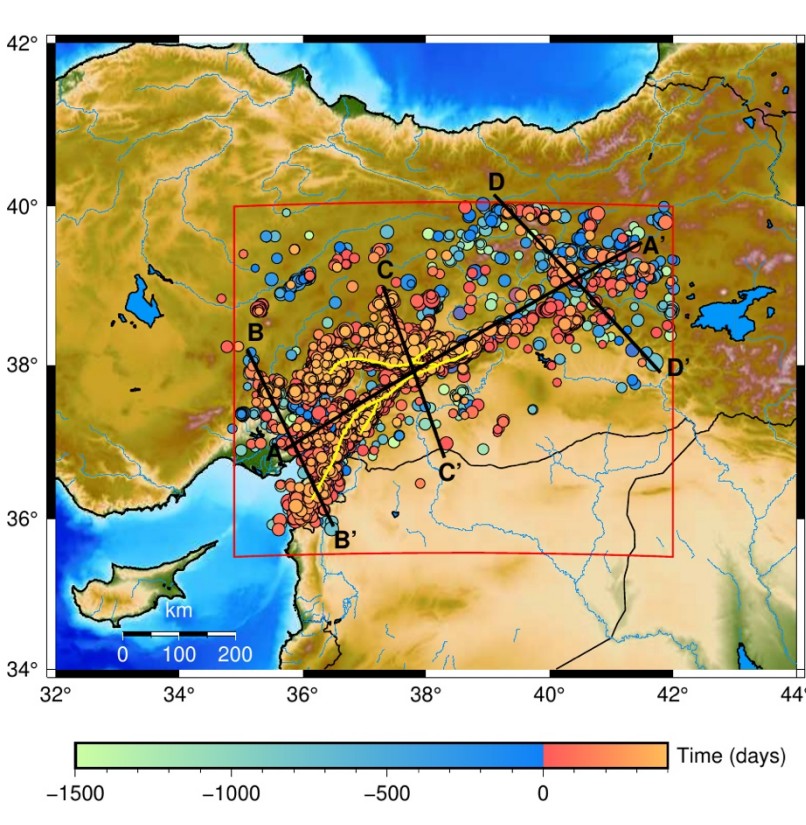


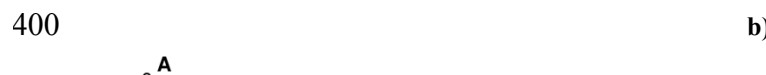

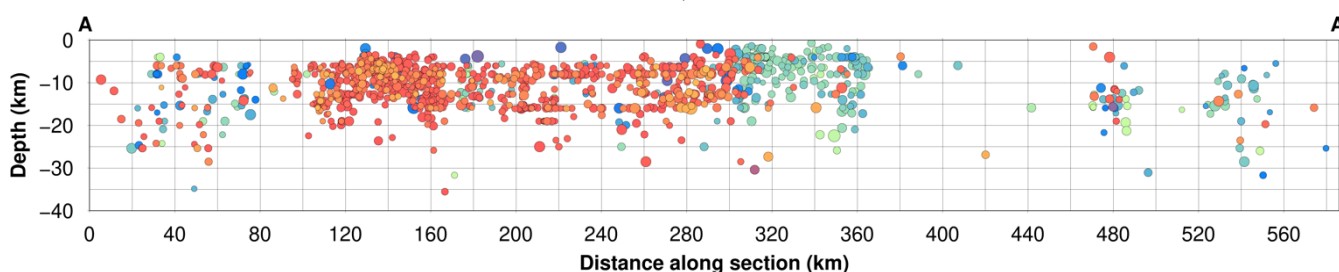

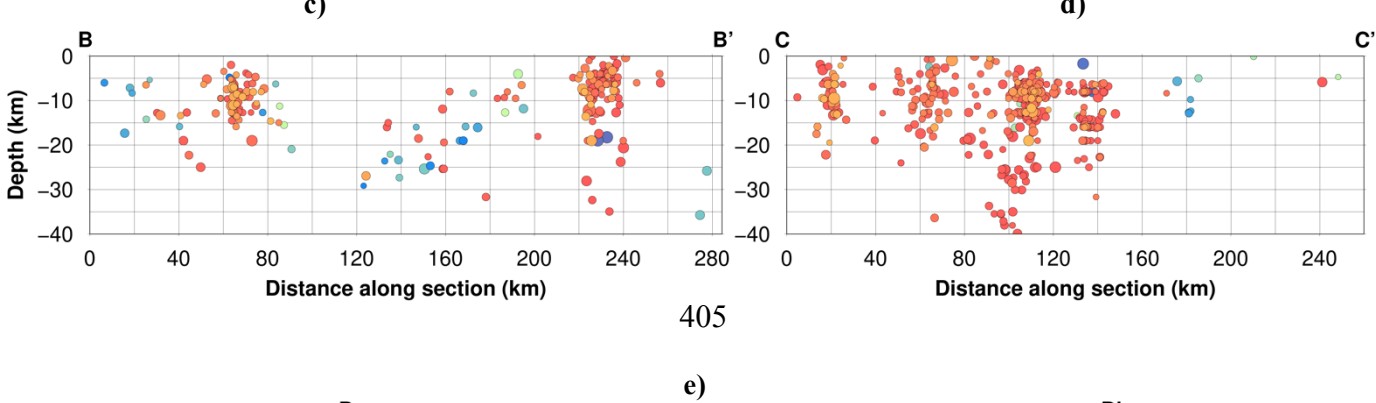


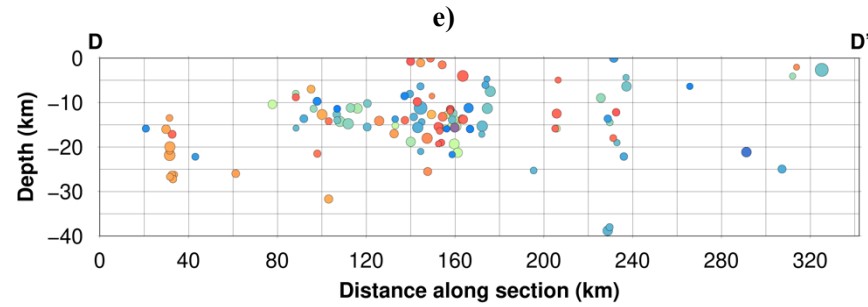

**Figure 14: a)** Seismicity map of the distributed catalog. The color scale represents the temporal evolution of the seismicity, while the size of the point is proportional to the magnitude. In red is shown the rectangle surrounding the study area, in yellow we show the faults according to Melgar et al. (2023). Profiles are shown with black lines.
**b)** Section A-A', length 586 km, 1303 events. **c)** Section B-B', length 284 km, 202 events. **d)** Section C-C'; length 260 km, 433 events. **e)** Section D-D'; length 342 km, 118 events.

Each profile was performed considering the events 10 km further to the left and 10 km to the right with respect to the line track. Vertical exaggeration of the cross-sections is 2:1.

Section A-A' (**Fig. 14b**) extends from southwest to northeast, running parallel to the average strike of the faults that generated

the two main earthquakes. In the first 80 km of the section, the seismicity appears to be scattered, while beyond 100 km, a

clear cluster emerges in the hypocentral area of Nurdağı. Here, the aftershocks are mainly concentrated at shallow depths (5-

18 km), which is consistent with the results of previous investigations (e.g. Melgar et al., 2023; Rodríguez-Pérez and Zúñiga, 2024).

Beyond 300 km along the profile, the aftershocks become much more sporadic. The first notable cluster corresponds to the Mw 6.7 Elazığ earthquake, which occurred on January 24, 2020 at 17:55 UTC near the town of Sivrice (Taymaz et al., 2021).

Our catalog includes the largest aftershock from this sequence (a Mw 5.1 event that occurred on January 25, 2020) along with 17 other events of Mw 4.0 or higher, for a total of about 100 events in the 40 days following the mainshock. In the northeasternmost part of the section, we observe a "seismicity gap" followed by an almost even distribution of foreshocks and aftershocks at a depth of 10-30 km over the last 100 km. The presence of the seismic gap (also evident in Melgar et al., 2023) is solely due to the time frame of our catalog, as Karabulut et al. (2023) shows that this section of the EAFZ has also

experienced moderate-magnitude earthquakes. In particular, the 2010 Kovancılar earthquake ruptured approximately 30 km of the northeasternmost extent of the EAFZ (Tan et al., 2011). Additionally, the adjacent Palu segment (~80 km long) partially ruptured between 2010 and 2011 producing two moderate-magnitude earthquakes (Mw 5.4 and Mw 6.1) and has remained continuously active.

Section B-B' (**Fig. 14c**) runs along the southwestern part of our study area: in its northern section, we find a cluster associated

with the second major event of 6 February 2023. As it progresses, the seismicity becomes more scattered at depths between 10 and 30 km before culminating in a well-defined cluster that extends over 20 km and is associated with aftershocks along the Nurdag-Pazarcık Fault.

Section C-C' (**Fig. 14d**), which is located near the center of our study area, is particularly significant as it transversely intersects the A-A' trace and the two main faults: the Nurdag-Pazarcık Fault to the North and the Sürgü Fault to the South. Aftershocks

are clearly visible along both faults, increasing in depth from North to South and reaching from the surface to a depth of about 40 km. In the last 100 km of the section, seismic activity is minimal, which is remarkable as this area is largely flat.

Finally, section D-D' (**Fig. 14e**), which focuses on the northeastern part of the study area, shows diffuse seismicity at different depths, including both very shallow and deep events, as also noted by Güvercin (2023). The temporal evolution of seismicity in this section shows an almost even distribution of foreshocks and aftershocks.

In all these considerations, it should not be forgotten that the seismotectonic setting in the region is ruled by a complex fault network that accommodates the stress generated by the relative motion at the triple junction of the Anatolian, Arabian, and African plates (Güvercin et al. 2022). Therefore, while these considerations provide a general overview of seismicity, a more detailed understanding requires in-depth studies of the mechanical behavior of the different EAF segments (Palo and Zollo, 2024; Wang and Barbot, 2024).


## 5 Data availability

The products derived by the procedures discussed above are available at the Zenodo repository:
https://doi.org/10.5281/zenodo.138389932 (Colavitti et al., 2024) in the form of tables, relative to events, stations, PGA, PGV and FAS. Open-access policy on these data has been adopted under the license CC BY 4.0. In this section we thus provide a description of the tables you can find in the repository, with the explanation of the fields relative to the presented dataset.

### 5.1 Earthquake file

The distributed dataset contains the following fields:

- **Id_AFAD** is the earthquake reference number according to AFAD
- **Date** is the event date (3 fields) in the format yyyy:mm:dd (years:months:days)
- **Time** is the origin time (3 fields) in the format hh:mm:ss (hours:minutes:seconds)
- **Ev_Lat** is the earthquake latitude in decimal degrees (°)
- **Ev_Lon** is the earthquake longitude in decimal degrees (°)
- **Ev_Depth** is the depth in kilometers (km)
- $M_L$ is the recalibrated local magnitude
- **StdM$_L$** is the $M_L$ standard deviation
- **Rms** is the root-mean-square error of local magnitude residuals at maximum likelihood or expectation hypocenter, expressed in seconds (s)
- **Erh** is the horizontal error in kilometers (km), given by NLLoc algorithm
- **Erz** is the vertical error in kilometers (km), given by NLLoc algorithm
- **Gap** is the maximum azimuth gap between stations used for location, expressed in decimal degrees (°)
- **Np** is the number of P-wave phases used for location
- **Ns** is the number of S-wave phases used for location
- **Ntot** is the total number of phases, both P and S-wave, used for location
- **Cov$_X$** is the covariance matrix value along X direction given by the NLLoc
- **Cov$_Y$** is the covariance matrix value along Y direction given by the NLLoc
- **Cov$_Z$** is the covariance matrix value along depth given by the NLLoc
- **Cov$_T$** is the covariance matrix related to the observed arrival times by the NLLoc

**5.2 Station file**

The distributed station file contains the following fields:

- **FDSN_Sta_Code** is the combined string code based on Network - Station - Location - Channel
- **Sta_Lat** is the latitude of the station in decimal degrees (°)
- **Sta_Lon** is the longitude of the station in decimal degrees (°)
- **Sta_Elev** is the elevation of the station in meters (m)

**5.3 PGA and PGV file**

The distributed PGA and PGV file contains the following fields:

- **Id_AFAD** is the earthquake reference number according to AFAD
- **FDSN_Sta_Code** is the combined string code based on Network - Station - Location - Channel
- **Dist_Hypo** is the hypocentral distance event-station in kilometers (km)
- **PGV_Z** is the Peak Ground Velocity relative to the vertical component (cm/s)
- **PGV_NS** is the Peak Ground Velocity in the N-S direction (cm/s)
- **PGV_EW** is the Peak Ground Velocity in the E-W direction (cm/s)
- **PGA_Z** is the Peak Ground Acceleration relative to the vertical component (cm/s²)
- **PGA_NS** is the Peak Ground Acceleration in the N-S direction (cm/s²)
- **PGA_EW** is the Peak Ground Acceleration in the E-W direction (cm/s²)

**5.4 FAS file**

We also report the acceleration Fourier Amplitude Spectra (FAS), for 98 frequency values from 0.05 to 47.2 Hz, equally spaced on the logarithmic scale.
Each file contains the following fields:

- **Id_AFAD** is the AFAD catalog reference ID
- **Ev_Lat** is the earthquake latitude in decimal degrees (°)
- **Ev_Lon** is the earthquake longitude in decimal degrees (°)
- **Depth** is the hypocentral depth in kilometers (km)
- **$M_L$** is the recalculated local magnitude
- **FDSN_Sta_Code** is the combined string code based on Network - Station - Location - Channel
- **Sta_Lat** is the station latitude in decimal degrees (°)

- **Sta_Lon** is the station longitude in decimal degrees (°)
- **Sta_Elev** is the elevation of the station in meters (m)
- **Dist_Hypo** is the hypocentral distance event-station in kilometers (km)
- **FAS_xxx_Z** is the acceleration FAS at xxx Hz relative to the vertical component (cm/s)
- **FAS_xxx_NS** is the acceleration FAS at xxx Hz in the N-S direction (cm/s)
- **FAS_xxx_EW** is the acceleration FAS at xxx Hz the E-W direction (cm/s)

Where **xxx** refers to a frequency between 0.05 and 47.2 Hz.

The frequencies used (in Hz) are as follows: 0.05, 0.08, 0.10, 0.13, 0.15, 0.17, 0.20, 0.22, 0.25, 0.28, 0.30, 0.32, 0.35, 0.38, 0.40, 0.43, 0.45, 0.47, 0.50, 0.53, 0.56, 0.59, 0.63, 0.67, 0.71, 0.75, 0.79, 0.84, 0.89, 0.94, 1.00, 1.06, 1.12, 1.19, 1.26, 1.33, 1.41, 1.49, 1.58, 1.67, 1.77, 1.88, 1.99, 2.11, 2.23, 2.37, 2.51, 2.65, 2.81, 2.98, 3.15, 3.34, 3.54, 3.75, 3.97, 4.21, 4.46, 4.72, 5.00, 5.30, 5.61, 5.94, 6.29, 6.67, 7.06, 7.48, 7.92, 8.39, 8.89, 9.42, 9.98, 10.57, 11.19, 11.86, 12.56, 13.30, 14.09, 14.93, 15.81, 16.75, 17.74, 18.79, 19.91, 21.08, 22.33, 23.66, 25.06, 26.54, 28.12, 29.78, 31.55, 33.42, 35.40, 37.49, 39.72, 42.07, 44.56 and 47.20.

## 6 Applications and Prospects

The 2019-2024 EAFZ high-quality data set offers numerous potential application and prospects. As we mentioned above, one of its main applications is spectral decomposition using the Generalized Inversion Technique (GIT), first introduced by Andrews (1986), Iwata and Irikura (1988) and Castro et al. (1990). The GIT is a reliable approach for the simultaneous investigation of source, path and site contributions to the observed ground motions in the frequency domain and plays a crucial role in improving the understanding of seismic processes and earthquake hazard assessment. This method is based on linear and time-invariant assumptions, for which the output is given by the convolution between the input with the transfer function of the system (Bindi et al., 2023b).

As we can see from **Fig. 15**, which shows the coverage map of the data set at 1 Hz, the study area is well sampled and dense of rays, especially along the main tectonic alignments of the EAF.

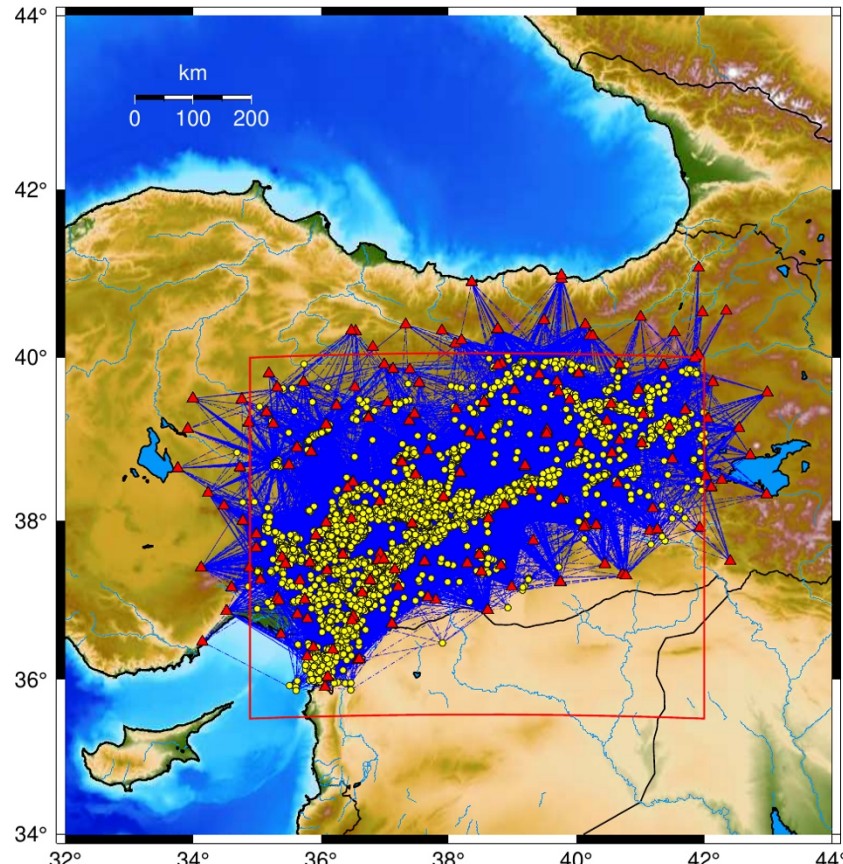

**Figure 15: Ray coverage map at f=1 Hz. Events are represented by yellow dots, stations by red triangles and rays by blue lines.**

One of the long-term objectives of the work is to provide a solid basis for the study of source parameters, similar to the efforts of the Southern California Earthquake Center (SCEC) community. There, spectral decomposition was applied to isolate source spectra of events belonging to the 2019 Ridgecrest seismic sequence (Bindi et al., 2023b), with a comparison study providing an epistemic analysis of the results uncertainties (Bindi et al., 2023c). From a spatial perspective, the data set provides excellent sampling across various frequencies (see **Figs. S11** and **S13** in the Supplement).

Ensuring comprehensive ray coverage is essential to obtain robust and meaningful seismological results, and is crucial to the success of GIT studies as it directly impacts the accuracy and reliability of the derived source, path and site parameters. Such dense ray coverage can also benefit the tomography community, especially for application such as Local Earthquake Tomography (LET), which uses first arrival times (Gokalp 2012; Ozer et al., 2019; Medved et al., 2021; Güvercin, 2023), or attenuation tomography studies (Koulakov et al., 2010; Toker and Şakir, 2022) in the EAFZ and surrounding regions. In this

sense, the disseminated data set is highly valuable, both in terms of the amount of data, and the quality of earthquake location and dense frequency sampling, which can help us improve the mapping of seismic structures in such a complex geological area.

The presented data set can be a valuable resource for the development of the STATION (Seismic sTATion and sIte amplificatiON, web page: https://distav.unige.it/rsni/station.php) service (Tarchini et al., 2024), which is a product based on the exchange and dissemination of seismological data from seismic stations in Italy and neighbouring regions. Starting from an automatic picking procedure of P and S phases, STATION guarantees a quasi-automatic elaboration of a selection of data records and is finalized to the calculation of Horizontal-to-Vertical Spectral Ratios (HVSR) and station specific $M_L$ residuals. A similar procedure has already been started for the EAFZ using this data set (see **Fig. S13**), with the aim of a precise site characterization.

In addition, the disseminated data set can significantly contribute to the development of the existing Ground Motion Prediction Equations (GMPEs) in the EAFZ and thus, to some extent to the improvement of earthquake hazard assessment (Akkar and Çağnan, 2010; Kale et al., 2015).

With over five years of recorded seismicity, the data set also enables the study of variations in the Q parameter which quantifies the attenuation of seismic energy through coda waves (Sertçelik, 2012) or, moving to higher frequencies, to support the investigation of the kappa ($\kappa$) parameter, which depends on the geological characteristics of the sites (Biro, 2024). In this context, recent studies in Central Italy (Castro et al., 2022b; Castro et al., 2025) based on high quality data sets that include low-to-moderate seismicity (Spallarossa et al., 2022) have shown that the temporal study of $\kappa$ can provide insights into the role of fluid circulation and contribute to the monitoring of seismic cycle.

One of the ultimate goals for which the data set was developed is to analyze the spatio-temporal evolution of seismicity and to investigate whether a preparatory process for preceded the February 6, 2023, Mw 7.8 earthquake (see Kwiatek et al., 2023; Picozzi et al., 2023b) and to understand how the identification of microseismicity is crucial for the detection and triggering of major events.

In the last few years, a set of physics-based features aimed at intercepting the preparatory phase of strong earthquakes have been developed (Picozzi et al., 2022; Picozzi et al., 2023a). In general, seismic sequences are analyzed based on the time-space intervals between earthquakes (Zaliapin et al., 2008; Zaliapin et al., 2016), which serve as an important tool for identifying seismic crises (see **Fig. S14**).

For a more detailed investigation of the spatio-temporal evolution, a future study will focus on analyzing the systematic deviations of the PGAs generated by each earthquake from the values predicted by a reference GMM calibrated for background seismicity, referred to as event-specific ground motion anomalies (eGMSs), as shown in Picozzi et al. (2024).

Finally, the data set can be effectively integrated with geodetic investigations such as Interferometric Synthetic Aperture Radar (InSAR), which provides pre-, co- and post- seismic deformation images and offer valuable insights into slow-slip events and fault behavior (for EAF, see An et al., 2023; He et al., 2023).

By integrating InSAR data with seismological information, it is indeed possible to gain a more comprehensive understanding of active tectonic mechanisms and fault dynamics in the region, which improves which ultimately improves earthquake processes analysis and seismic risk assessment.

# 7 Conclusive remarks

This work presents a seismic catalog that covers the period between January 1, 2019 and February 29, 2024 and includes both the pre- and post-seismic periods of the devastating February 6, 2023 earthquake sequence that struck southern and central Türkiye and northern and western Syria along the EAFZ. The data set focuses on small-to-moderate earthquakes in the $M_L$ range 2.0-5.5 and is intended as a valuable tool for researchers investigating seismic source characterization and strong motion parameters.

The high-quality catalog of this was achieved with the application of the CASP (Scafidi et al., 2019) software, which allowed the identification of 270,704 seismic phases (148,223 P- and 122,481 S-wave first arrivals) for a total of 9,442 events recorded by 271 stations. All events were located with the NLLoc algorithm (Lomax et al., 2000; Lomax et al., 2012). An initial velocity model specifically suited to the EAFZ was used, resulting in reliable earthquake locations with an uncertainty of ±1 km for both horizontal and depth location. Notably, our depth estimates differ from those of the AFAD reference catalog and appear to be consistent with the geometrical complexities of the fault segments involved in the 2023 Türkiye earthquake sequence (Palo and Zollo, 2024). In addition, the distributed catalog contains $M_L$ values calibrated using a non-parametric approach (Bindi et al., 2020).

The last section of this paper deals with possible applications of the data set. It was developed specifically for spectral decomposition, allowing for the separation and analysis of key factors such as source characteristics, attenuation and site effects. In addition, the new event locations can support research on attenuation in terms of Q factor or $\kappa$ parameter. An important long-term goals of the catalog is to understand the spatio-temporal evolution of seismicity, identifying potential proxies to intercept the preparatory phase of strong earthquakes (Picozzi et al., 2023b).

We strongly believe that the creation of high-quality, standardized, and open-source seismic data sets, including FAS and widely used strong motion parameters, such as PGA and PGV, is essential for the advancement of seismological and earthquake engineering research.

## Code availability

Most of the figures were generated using MatLab software (https://mathworks.com/products/matlab.html, MathWorks, 2023). We used the Generic Mapping Tools (https://www.generic-mapping-tools.org/; Wessel et al., 2013) to produce **Figs. 1, 14** and **15**. **Figure 12** is done through the ObsPy package (https://docs.obspy.org/; Beyreuther et al., 2010), a Python framework for processing seismological data. Seismic waveforms, pick observations and P and S travel times are available on request by contacting the author. The data set is freely available at the Zenodo repository: https://doi.org/10.5281/zenodo.13838992 (Colavitti et al., 2024) under the license CC BY 4.0.

## Author contribution

DSp, DB and MP conceptualized the study. DSp developed the code used to compile the disseminated data set, DB recalibrate the local magnitude. LC developed the quality checks; LC realized the images of the manuscript, with the contribution of DSc and GT.
LC organized the publication and, with the contribution of DB, GT, DSc, MP and DSp wrote the first draft of the manuscript. All authors participated to the finalization of the article.

## Competing interests

The contact author has declared that none of the authors has any competing interests.

## Acknowledgements

The authors are very grateful to the handling topic editor Dr. Andrea Rovida and two anonymous reviewers for their valuable comments that improved this manuscript.

## Financial support

This research is supported by the Project PREPARED "PREparatory Phase of lARge earthquakes from seismic information and gEodetic Displacement", project code 2022ZHWC9 in the frame of Bando PRIN 2022 - Progetto di Rilevante Interesse Nazionale (D. D. MUR n. 746, 31-05-2023).

## Review statement

This paper was edited by Andrea Rovida and reviewed by two anonymous referees.

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
