# Peer review of "A high-quality Data Set for seismological studies in the East Anatolian Fault Zone, Türkiye"

_Earth System Science Data, 2024_

## Referee Comment (RC2)

The paper by Leonardo Colavitti et al. presents a new seismic catalog for the East Anatolian fault zone. The data span is limited to 5 years and 2 months, and the authors provide a robust collection of strong motion parameters and a refined attenuation function for local magnitude calculations. This work and its deliverables are undoubtedly an asset for the seismological community studying the region, and it seems the consistency checks—which are mandatory for automatic catalogs—were properly implemented.

However, the paper could benefit from additional insights to enhance its already good quality. Overall, I found the paper to be well written in English, though there is room for improvement, particularly in the structure and arrangement of the text. I would recommend publication after addressing the following comments.

**MAIN**
- In the introduction, the authors mention that they have excluded the two main shocks (7.8 and 7.6 Mw), along with 22 other events with a magnitude greater than 5.6 (Ml > 5.6), from this catalog because these events have already been published in other works (P3L60). I would like to know whether these events were completely overlooked during the processing or if they were intentionally excluded afterward. If they were excluded, and calculations for these events have already been performed, I believe that the results should be made accessible to the community.
- I was surprised not seeing any cross-sections showing the seismicity at depth. The authors should consider adding some of these cross-sections around the main fault structures in the investigation region.
- In addition, the author should provide a minimum discussion on the observed seismicity in their catalogue with previous studies. For example, the entire section 4 could be moved to the supplementary and replaced by such a discussion section. What about the deepest events shown at the border of the investigation area in SM1: Are those depths expected, or are they somehow biased by the methods applied?
- What is the magnitude of completeness of this catalog? Did the authors consider conducting a magnitude completeness study? This additional study would enhance the already good quality of the manuscript.

**FIGURES:**
- *All the figures showing a histogram distribution should also include a text box stating the main statistics (count, mean, median, std, mad). This applies to both the main manuscript and the supplementary material.*
- Figure 10 should display a scatter plot comparing the new magnitude of local events (ML) with the corresponding reference magnitude (ML ref.) derived from the initial AFAD catalog, rather than presenting two separate histogram distributions. The author might consider using different sub-panels for clarity if multiple magnitudes are included.
- The supplementary figures S5 to S7 should be grouped to improve information access.
- In my opinion, the information in Figure SM9 is not properly presented. Displaying the station corrections on a map would be a better choice.

- Figure SM10 can be displayed with an equal fixed limit for both X and Y axis to enhance readability.

**MINOR**

- The author briefly outlined the main characteristics of the procedure for calculating the FAS. A similar description should be provided for the RSNI picker. Please include bullet points highlighting the key steps and the pickers used, potentially in section 3.1. Additionally, clarify how the phase-filtering quality check was conducted. What is the absolute timing associated with the weighting class?
- P4L84: specify the initial magnitude range for event-selection.
- P5L93: Please specify the last time the website was accessed.
- What will happen to the final pick observations listed in the catalog? Will they be released publicly, or are they already available? Please add one sentence about this to the Data Availability section.
- In the manuscript, the authors state their catalog range "…between 2.0 and 5.5 Ml", but the distributed CSV file shows a minimum ML of 1.87. I suggest updating that information.
- The 1D model utilized with NLL is derived from Guvercin (2023). Could you clarify whether it is based on the Minimum 1D model or an extrapolation from the final 3D model? Additionally, the original 1D paper presents significantly more than 12 layers and indicates an average Vp/Vs ratio of 1.74, while your SM4 table shows a ratio of 1.73. I believe that neither of these discrepancies has a significant effect on the study. However, it would be beneficial to specify the actual modifications made in the text for clarity.
- I suggest double-checking and modifying some English structures to improve reading fluency. Like passive forms (i.e. on P5L91), or section 4 titles: "File of earthquakes" → "Earthquakes file", "File of stations" → "Stations file" etc …

---

## Author Comment (AC1)

**Revisor #1**

**The ms describes the features of the accelrometric data in Eastern Turkiye. Authors had compiled low magnitude events in their dataset. The paper is well-written and can be accepted however some minor revisions are required.**

Thank you for your time and willingness to read our manuscript. You can find the requested additions highlighted in yellow in the new version of the manuscript.

**Section3.1.1: What is the definition of "error"?**

In this work, we can distinguish 3 types of error, derived from the outputs of the NLLoc (Non Linear Location) algorithm (see Lomax *et al.*, 2000). These errors are:

- Horizontal error;
- Depth error;
- Root Mean Square (RMS) error.

In the NNLoc algorithm, the errors in the observations are assumed to be Gaussian.

The horizontal error quantifies the uncertainty in the earthquake location within the horizontal plane since is the projection of the ellipsoid onto the horizontal plane, typically represented as the semi-major and semi-minor axes of the projection. This error is expressed as a confidence region in the horizontal plane, indicating the area where the earthquake is likely located with a specified probability.

The depth error measures the uncertainty in the vertical position of the earthquake hypocenter and is taken directly from the vertical axis of the error ellipsoid. Both horizontal and depth errors are affected by several factors, as geometry of the seismic network, accuracy of travel-time measurements and complexity of the velocity model.

Finally, the RMS error is a measure of the fit between observed and computed travel times for the final earthquake location, as expressed in the Eq. 1 of the manuscript.

All these outputs combined allow seismologists to assess the reliability of the earthquake location in terms of both spatial precision and the quality of the model-data fit.

We have added some of these considerations in section 3.1.1 of the revised version of the manuscript.

**Could you plot the difference between your estimation and AFAD's in terms of distance between epicenters (e.g. depth was compared in Section 3.1.2).**

The AFAD catalog contains any information on the location error and unfortunately makes reproducibility impossible. What we can do instead is to plot the empirical Cumulative Distribution Function (CDF) that directly compares the location of each event, computing the distance between the location obtained in this work and the one provided by the AFAD catalog, as shown in **Fig. 1_Rev#1.** The distance is computed through the haversine formula (see Robusto, 1957), that determines the great circle distance between 2 points on a sphere given their longitudes and latitudes.

[Figure]

**Figure 1_Rev#1: Cumulative Distribution Function (CDF) with the Haversine distance computed between the location obtained in this work and the one provided by the AFAD catalog. Black dashed line shows the distance at 10 km, corresponding to the 92.7% of the CDF (blue curve); red point shows the intercept.**

As we can see from the Figure above, the CDF shows that about 80% of the entire data set has a difference in distance in the catalogs of 5 km, 92.7% are below 10 km (the intercept is represented by the red point) and almost the entire data set (above 97%) below 20 km. This information is a further indication that our catalog has reliable and precise epicenters.

We add this image (Figure 7) and some considerations in the subsection *3.1.2 Epicentral comparison with AFAD catalog* in the revised version of the manuscript.

**Figure 7&10: I think side-by-side bars would be more readable instead of overlapping bars.**

Ok, thanks for your suggestion. We modified Figures 7 and 10 (now Figures 8 and 11) using two separated figures compared to one figure with overlapping bars.

**References**

AFAD: Disaster and Emergency Management Presidency. National Seismic Network of Turkey (DDA), International Federation of Digital Seismograph Networks available at http://tdvm.afad.gov.tr/ (last accessed February 2024).

Lomax, A., Virieux, J., Volant, P., and Berge-Thierry, C.: Probabilistic earthquake location in 3D and layered models: Introduction of a Metropolis-Gibbs method and comparison with linear locations, in Adv. in Seismic Event Location, eds. Thurber, C. H. and Rabinowitz, 101-134, Kluwer Academic Publishers, 2000.

Robusto, C. C.: The cosine-haversine formula. The American Mathematical Monthly, 64(1), 38-40, 1957.

Best Regards,

Leonardo Colavitti, on behalf of the authors.
* * *
Leonardo Colavitti, PhD
Post Doctoral Research Fellow

University of Genoa (Italy)
DISTAV (Department of Earth, Environment and Life Sciences)
Seismology Lab of RSNI (Regional Seismic Network of Northwestern Italy)

---

## Author Comment (AC2)

**Revisor #2**

**The paper by Leonardo Colavitti et al. presents a new seismic catalog for the East Anatolian fault zone. The data span is limited to 5 years and 2 months, and the authors provide a robust collection of strong motion parameters and a refined attenuation function for local magnitude calculations. This work and its deliverables are undoubtedly an asset for the seismological community studying the region, and it seems the consistency checks - which are mandatory for automatic catalogs - were properly implemented.**

**However, the paper could benefit from additional insights to enhance its already good quality. Overall, I found the paper to be well written in English, though there is room for improvement, particularly in the structure and arrangement of the text. I would recommend publication after addressing the following comments.**

Thank you for reading the manuscript and for your precious suggestions, which were helpful in improving the manuscript. You can find the requested additions highlighted in green in the new version of the paper.

**MAIN**

**- In the introduction, the authors mention that they have excluded the two main shocks (7.8 and 7.6 Mw), along with 22 other events with a magnitude greater than 5.6 (Ml > 5.6), from this catalog because these events have already been published in other works (P3L60). I would like to know whether these events were completely overlooked during the processing or if they were intentionally excluded afterward. If they were excluded, and calculations for these events have already been performed, I believe that the results should be made accessible to the community.**

The automatic processing procedure used in this study has also been applied to these events; however, the results were intentionally excluded afterward. The main reason is that our objective was to disseminate a data set of small and moderate events processed homogeneously using an automatic procedure. Large events require more *ad-hoc* processing, which has already been applied in the aforementioned strong-motion databases (e.g. ESM – Engineering Strong Motion Database (Luzi et al., 2020), see https://esm-db.eu/#/home), making them readily available and easily accessible.

Moreover, for the application of the Generalized Inversion Technique (GIT) decomposition (Andrews, 1986; Castro et al., 1990) to isolate source, propagation, and site effects, and subsequently evaluate source parameters, large events do not meet the point-source assumptions (Brune, 1970) and introduce complications related to the far-field approximation.

**- I was surprised not seeing any cross-sections showing the seismicity at depth. The authors should consider adding some of these cross-sections around the main fault structures in the investigation region.**

Thanks for the suggestion. We have added the Figure you can find below in the manuscript (**Fig. 14** in the new version of the manuscript) representing the catalog map to be distributed and where the color scale represents the temporal evolution of seismicity, while the point size is proportional to the magnitude.

**a)**

[Figure]

**b)**

[Figure]

[Figure]

**Figure 1_Rev#2:**

a) Seismicity map of the distributed catalog. The color scale represents the temporal evolution of the seismicity, while the size of the point is proportional to the magnitude. In red is shown the rectangle surrounding the study area, in yellow we show the faults according to Melgar et al. (2023). Profiles are shown with black lines.

b) Section A-A', length 586 km, 1303 events. **c)** Section B-B', length 284 km, 202 events. **d)** Section C-C'; length 260 km, 433 events. **e)** Section D-D'; length 342 km, 118 events.
Each profile was performed considering the events 10 km further to the left and 10 km to the right with respect to the line track. Vertical exaggeration of the cross-sections is 2:1.

**In addition, the author should provide a minimum discussion on the observed seismicity in their catalogue with previous studies. For example, the entire section 4 could be moved to the supplementary and replaced by such a discussion section.**

In the new version of the manuscript, we have introduced a new section called "Discussions: Seismicity distribution in the EAFZ" where we have provided a discussion of the evolution of the spatio-temporal seismicity of this catalog by comparing it with other works (e.g. Melgar et al., 2023)

As requested, we have also moved the previous section 4 concerning the construction of the final catalog to the supplementary material (now called **SM13**).

**What about the deepest events shown at the border of the investigation area in SM1: Are those depths expected, or are they somehow biased by the methods applied?**

The deeper events shown at the border of the investigation area reflect an area where deeper crustal roots are recorded (as is also visible from the cross-sections above) and are therefore reliable results.

**- What is the magnitude of completeness of this catalog? Did the authors consider conducting a magnitude completeness study? This additional study would enhance the already good quality of the manuscript.**

As requested, we have estimated the b-value and the b-positive for all the 78,728 events downloaded from the AFAD service and the 9,442 earthquakes used for the GIT study.

However, it is worth noting that the GIT data sets considered was generated with the purpose of performing source parameters studies, and thus has been guided by a different level of event selection with respect to the ALL one, with the aim of considering only high quality recordings among all those variable with the origin database. Even considering the incomplete nature of our GIT data set, we have analyzed its cumulative frequency-magnitude (CFM) distribution adopting standard approaches used in seismic hazard as done for the ALL data set.

For the b-value, we have applied the entire-magnitude-range method (Woessner and Wiemer, 2005), which is implemented in the software package ZMAP (Wiemer, 2001). This software allows the simultaneous estimation of the completeness magnitude Mc and of the parameters $a$ and $b$ of the Gutenberg-Richter law (Gutenberg and Richter, 1944).

In particular, the b-value, is obtained by the maximum likelihood approach (Aki, 1965) while the uncertainty of the b-value is computed by means of a bootstrap approach (Efron, 1979). For each data set, we perform 200 realizations of random sampling with replacement depending on the analysis. We have also estimated the b-positive applying the procedure proposed by van der Elst (2021). For both parameters and data sets, we have varied the low magnitude cutoff to investigate the robustness of the estimates.

ALL Data set

**Figure S1a_Rev#2** shows the frequency-magnitude distribution (bin 0.1) and the Gutenberg-Richter law colored for different low magnitude cutoff. As shown in Figure **S1c_Rev#2**, the b-value stabilizes at ≈ - 0.9 for Mc ≥ 2. Figures S1b and S1c show similar results but for the b-positive. Here, in agreement with van der Elst (2021), we observed that b-positive is much less affected than the b-value and b-positive estimates for different low magnitude cutoff values (i.e., the probability is defined as $e^{-misfit}$, where the misfit is L$_2$ norm computed considering the CFM and the Gutenberg-Richter law). The best estimates are b-value $0.89 \pm 0.01$ and b-positive $0.85 \pm 0.01$.

GIT Data set

**Figure S2_Rev#2** shows, with the same scheme of **Fig.S1_Rev#2**, the results of the analyses for the GIT data set. Also in this case, we observe that the b-positive (**Fig. S2d_Rev#2**) is more robust than the b-value (**Fig. S2c_Rev#2**). Nevertheless, the best-fit estimates agree, being both the b-value and b-positive equal to $0.9 \pm 0.01$.

[Figure]

**Figure 2_Rev#2:**
**a) Cumulative frequency magnitude distribution for the ALL data set and the Gutenberg-Richter law colored for different low magnitude cutoff.**
**b) The same as a), but for b-positive.**
**c) b-value estimates for different low magnitude cutoff.**
**d) The same as c), but for b-positive.**
**e) Probability of b-value and b-positive estimates.**

[Figure]

**Figure 3_Rev#2:**
**a) Cumulative frequency magnitude distribution for the GIT data set and the Gutenberg-Richter law colored for different low magnitude cutoff.**
**b) the same as a), but for b-positive.**
**c) b-value estimates for different low magnitude cutoff.**
**d) the same as c), but for b-positive.**
**e) probability of b-value and b-positive estimates.**

We added some of these information in the **Supplementary Material SM10** and **SM11** and at the end of the Section "Magnitude Computation".

**FIGURES:**

**- All the figures showing a histogram distribution should also include a text box stating the main statistics (count, mean, median, std, mad). This applies to both the main manuscript and the supplementary material.**

We have updated all histograms in both the manuscript and supplementary material by adding a box in the top right-hand corner with the main statistics: number of events or records, mean value, median, standard deviation (SD) and mean absolute deviation (MAD).

We have therefore updated the following images (the number regards the new version of the manuscript):

- **Fig. 5b)** and **c)**
- **Fig. 6b)** and **c)**
- **Fig. 8 a)** and **b)**
- **Fig. 10 a)** and **b)**
- **Fig. 11 a)** and **b)**
- **SM 5 a)**, **b)** and **c)**

For histograms relating to the number of P-phases, or the number of S-phases or the total number of phases, we chose to use an integer for the median and the mean.

**- Figure 10 should display a scatter plot comparing the new magnitude of local events (ML) with the corresponding reference magnitude (ML ref.) derived from the initial AFAD catalog, rather than presenting two separate histogram distributions. The author might consider using different sub-panels for clarity if multiple magnitudes are included.**

Thanks for the suggestion. We have replaced **Fig. 10** (now **Figure 11**) by using 2 separate figures for greater clarity, al also requested by the Reviewer #1.

**- The supplementary figures S5 to S7 should be grouped to improve information access.**

We have grouped the figures of supplementary material **S5**, **S6** and **S7** into a single figure (**SM5**) and changed the references within the text accordingly.

**- In my opinion, the information in Figure SM9 is not properly presented. Displaying the station corrections on a map would be a better choice.**

Thanks for the suggestion; we have included the map with the station corrections in Fig. **SM9a** (now **SM7a**) coloring the stations using the scale from blue (negative values), white (neutral values) and red (positive values).

**- Figure SM10 can be displayed with an equal fixed limit for both X and Y axis to enhance readability.**

We have used the same scale along X and Y in **Figure SM10** (now **SM8**) which is shown below to facilitate readability.

[Figure]

**Figure 4_Rev#2:**
**SM10: Comparison between non parametric magnitude (MNP) with recomputed local magnitude M$_L$ (a) and local magnitude from AFAD catalogue (b). Dashed lines correspond to the line 1:1 shifted by $\pm$ 2 standard deviations of the residuals between the reported magnitudes. Value of 1 Standard deviation: a) 0.13; b) 0.25.**

**MINOR**

**- The author briefly outlined the main characteristics of the procedure for calculating the FAS. A similar description should be provided for the RSNI picker. Please include bullet points highlighting the key steps and the pickers used, potentially in section 3.1.**

Thank you for your suggestion. We have included a brief description of the key modules of the CASP (including the RSNI picker) at the beginning of the section 3.1, as suggested. A more detailed description of CASP can be retrieved in the works of Spallarossa et al. (2014), Scafidi et al. (2016), Scafidi et al. (2018) and Scafidi et al. (2019).

**Additionally, clarify how the phase-filtering quality check was conducted. What is the absolute timing associated with the weighting class?**

The RSNI-picker includes a weighting mechanism and quality classification rigorously calibrated on a series of reference picks, with corresponding observation error estimates provided by an expert analysis. The quality weighting mechanism can be tuned toward predicting the same quality classifications as those that would be estimated by the user.

Following the strategy of Aldersons (2004), the RSNI-picker adopts a weighting mechanism that separately assigns each automatic P- and S-phase pick to a quality class, on the basis of discriminating variables (predictors) derived from picking procedures. The RSNI-picker provides arrival times together with quality factors that indicate the estimated uncertainty of the automatic phases. These quality classes are those typically expected by many seismic codes to define the quality and reliability of each arrival time. Lower values (0 or 1) imply the best quality, whereas higher values (2 or 3) denote the poorest quality.

More details are available in Spallarossa et al. (2014).

**- P4L84: specify the initial magnitude range for event-selection.**

Done.

**- P5L93: Please specify the last time the website was accessed.**

Done, last access was done on 29 February 2024.

**- What will happen to the final pick observations listed in the catalog? Will they be released publicly, or are they already available? Please add one sentence about this to the Data Availability section.**

At this stage, we decided not to release publicly the final pick observations. However, seismic waveforms and P and S travel times are available on request by contacting the author. We have added this statement in the section "Code and data availability".

**- In the manuscript, the authors state their catalog range "…between 2.0 and 5.5 Ml", but the distributed CSV file shows a minimum ML of 1.87. I suggest updating that information.**

Thank you for the observation. The magnitudes we refer to are those of the events in the AFAD's reference catalog. The calibration of the local magnitude using the non-parametric approach may differ from the AFAD magnitude and for this reason the minimum value is 1.87 and not 2.

**- The 1D model utilized with NLL is derived from Güvercin (2023). Could you clarify whether it is based on the Minimum 1D model or an extrapolation from the final 3D model?**

The velocity model used in this study is taken from the work of Güvercin et al. (2022) "Active Seismotectonics of the East Anatolian Fault" published in Geophysical Journal International (we have actually updated the references in the manuscript as well, where we only cited Güvercin, 2022).

The model used for the NNLoc is a 1-D model and is the one reported in the Table **SM4** of the supplementary material of her paper, as shown in the table below.

| Depth (km) | Vp (km/s) | Vs (km/s) |
|---|---|---|
| 0 | 3.88 | 2.04 |
| 1 | 4.52 | 2.43 |
| 2 | 5.62 | 3.03 |
| 4 | 5.75 | 3.31 |
| 6 | 5.85 | 3.38 |
| 8 | 5.96 | 3.43 |
| 10 | 6.00 | 3.44 |
| 12 | 6.05 | 3.46 |
| 16 | 6.32 | 3.62 |
| 20 | 6.40 | 3.67 |
| 25 | 6.83 | 3.92 |
| 30 | 6.89 | 3.94 |
| 37 | 7.80 | 4.40 |
| 45 | 8.22 | 4.56 |
| 60 | 8.30 | 4.61 |

**Table 1_Rev#2: 1-D velocity model used by Güvercin et al. (2022).**

The model is obtained using 700 events using VELEST inversion code (Kissling, 1994). The selected events have azimuthal gap < 80° and are relocated using at least 25 phase readings.

**Additionally, the original 1D paper presents significantly more than 12 layers and indicates an average Vp/Vs ratio of 1.74, while your SM4 table shows a ratio of 1.73. I believe that neither of these discrepancies has a significant effect on the study. However, it would be beneficial to specify the actual modifications made in the text for clarity.**

Yes, the paper of Güvercin et al. (2022) shows 14 layers and a half-space: actually for the upper crust Vp/Vs ratio has values approximately equal to 1.73 and then up to Moho increasing values up to 1.74 and Vp/Vs values equal to 1.80 in the mantle. In the previous version of the manuscript, we had reported a simplified interpolated model, but the one we actually used is the one shown in the **Table S1_Rev#2**. We apologize for the inaccuracy.

We have added a few lines concerning the velocity model in the new version of the manuscript.

**- I suggest double-checking and modifying some English structures to improve reading fluency. Like passive forms (i.e. on P5L91), or section 4 titles: "File of earthquakes" à "Earthquakes file", "File of stations" à "Stations file" etc …**

Thanks for the suggestion. We have completely revised the manuscript to improve reading fluency.

**References**

AFAD: Disaster and Emergency Management Presidency. National Seismic Network of Turkey (DDA), International Federation of Digital Seismograph Networks available at http://tdvm.afad.gov.tr/ (last accessed February 2024).

Aki, K.: Maximum Likelihood Estimate of b in the Formula log10N=a-bM and its confidence limits. Bull. Earthq. Res., 43, 237-239.

Aldersons, F.: Toward a three-dimensional crustal structure of the Dead Sea region from local earthquake tomography, PhD thesis, Tel Aviv University, Israel, 120 pages. http://faldersons.net, 2004.

Andrews, D. J.: Objective determination of source parameters and similarity of earthquakes of different size. Earthquake source mechanics, 37, 25w9-267, https://doi.org/10.1029/GM037p0259, 1986.

Brune, J. N. Tectonic stress and the spectra of seismic shear waves from earthquakes, J. Geophys. Res., 100 (26), 4997-5009.

Castro, R. R., Anderson, J. G., Singh, S. K.: Site response, attenuation and source spectra of S waves along the Guerrero, Mexico, subduction zone, Bull. Seism. Soc. Am., 80 (6A), 1481-1503, https://doi.org/10.1785/BSSA08006A1481, 1990.

Efron, B.: Bootstrap Methods: Another Look at the jackknife, Ann. Stat. 7, 1-26, 1979.

Gutenberg, B. and Richter, C. F., 1944. Frequency of earthquakes in California. Bull. Seism. Soc. Am. 34(4), 185-188. doi: https://doi.org/10.1785/BSSA0340040185.

Güvercin, S. E., Karabulut, H., Konca, A. O., Doğan, U., and Ergintav, S.: Active seismotectonics of the East Anatolian Fault, Geophys. J. Int., 230: 50-69. https://doi.org/10.1093/gji/ggac045, 2022.

Kissling, E., Ellsworth, W. L., Eberhart-Phillips, D., and Kradolfer, U.: Initial reference models in local earthquake tomography, J. Geophys. Res.: Solid Earth, 99: 19635-19646. https://doi.org/10.1029/93JB03138, 1994.

Lomax, A., Virieux, J., Volant, P., and Berge-Thierry, C.: Probabilistic earthquake location in 3D and layered models: Introduction of a Metropolis-Gibbs method and comparison with linear locations, in Adv. in Seismic Event Location, eds. Thurber, C. H. and Rabinowitz, 101-134, Kluwer Academic Publishers, 2000.

Luzi, L., Lanzano, G., Felicetta, C., D'Amico M. C., Russo, E., Sgobba S., Pacor, F., and ORFEUS Working Group 5, Engineering Strong Motion Database (ESM) (Version 2.0), Istituto Nazionale di Geofisica e Vulcanologia (INGV), https://doi.org/10.13127/ESM.2, 2020.

Melgar, D., Taymaz, T., Ganas, A., Crowell, B. W., Öcalan, T., Kahraman, M., Tsironi, V., Yolsal-Çevikbilen, Valkaniotis, S., Irmak, T. S., Eken, T., Erman, C., Özkan, Doğan, A. H., Altuntaş, C.: Sub- and super-shear ruptures during the 2023 Mw 7.8 and Mw 7.6 earthquake doublet in SE Türkiye, Seismica 2, 3, https://doi.org/10.26443/seismica.v2i3.387, 2023.

Scafidi, D., Spallarossa, D., Tunino, C., Ferretti, G., Viganò, A.: Automatic P- and S-Wave Local Earthquake Tomography: Testing Performance of the Automatic Phase-Picker Engine "RSNI-Picker", Bull. Seismol. Soc. Am., 106(2), 526-536, https://doi.org/10.1785/0120150084, 2016.

Scafidi, D., Viganò, A., Ferretti, G., and Spallarossa, D.: Robust picking and accurate location with RSNI-Picker$_2$: Real-Time Automatic Monitoring of Earthquakes and Non Tectonic Events, Seism. Res. Lett., 89(4), 1478-1487, https://doi.org/10.1785/0220170206, 2018.

Scafidi, D., Spallarossa, D., Ferretti, G., Barani, S., Castello, B., and Margheriti, L.: A Complete Automatic Procedure to Compile Reliable Seismic Catalogs and Travel-Time and Strong-Motion Parameters Datasets, Seism. Res. Lett. 90(3), 1308-1317, https://doi.org/10.1785/0220180257, 2019.

Spallarossa, D., Ferretti, G., Scafidi, D., Turino, C., and Pasta, M.: Performance of the RSNI-Picker, Seismol. Res. Lett., 85(6), 1243-1254, https://doi.org/10.1785/0220130136, 2014.

Wiemer, S.: A Software Package to Analyze Seismicity: ZMAP. Seism. Res. Lett. 72(3), 373-382. doi: https://doi.org/10.1785/gssrl.72.3.3.373.

van der Elst, N. J.: B-Positive: A Robust Estimator of Aftershock Magnitude Distribution in Transiently Incomplete Catalogs. J. Geophys. Res.-Solid Earth 126, e2020JB021027. doi: https://doi.org/10.1029/2020JB021027.

Woessner, J., Wiemer S.: Assessing the Quality of Earthquake Catalogues: Estimating the Magnitude of Completeness and Its Uncertainty. Bull. Seism. Soc. Am. 95 (2), 684-698. doi: 10.1785/0120040007.

Best Regards,

Leonardo Colavitti, on behalf of the authors.
* * *
Leonardo Colavitti, PhD
Post Doctoral Research Fellow

University of Genoa (Italy)
DISTAV (Department of Earth, Environment and Life Sciences)
Seismology Lab of RSNI (Regional Seismic Network of Northwestern Italy)

---

## Author Response (AR2)

**Reply to Editor**

Dear Authors

Thank you for revising your manuscript and for the detailed responses to the reviews. Although I support the inclusion of the discussion on the seismicity distribution as suggested in Review #2, the request for moving the description of the dataset to the supplementary material does not match the requirements of a data paper, unfortunately.

I suggest you double-check ESSD's submission guidelines (https://www.earth-system-science-data.net/submission.html).

In particular, to accept the MS for publication, I recommend restoring the original section "Data Availability" as Section 5 (before the conclusions, as per the guidelines), which must also specify the license associated with the dataset.

For the track-change version of the revised manuscript, please use the corresponding function in your text editor.

We look forward to receiving your revised manuscript.

Best regards

Andrea Rovida
* * *
Dear Editor,

Thank you for the changes you have indicated to be made in the manuscript.

First of all, I have adapted the labelling of the supplementary material (as indicated by Katja Gänger) in agreement with the journal's guidelines, also updating and modifying the references within the manuscript accordingly.

Additionally, as you pointed out, I have restored the original "Data Availability" section, removing it from the supplement and placing it after the discussion section and before the conclusions (which I have renamed "Conclusive Remarks"). I have included the license under which the data is distributed (CC BY 4.0) both in the dedicated data section (now 5. Data Availability) and at the end of the "Code Availability" paragraph.

All changes in the manuscript, as well as in the supplements, are highlighted in yellow.

I hope I have addressed all the requests, and I remain available for any further clarifications.

Best regards,

Leonardo Colavitti, on behalf of the authors.